# Inhibitory activities of short linear motifs underlie Hox interactome specificity in vivo

**Manon Baëza[1,2†], Séverine Viala[1,2†], Marjorie Heim[1,2†], Amélie Dard[1,2], Bruno Hudry[3], Marilyne Duffraisse[1,2], Ana Rogulja-Ortmann[4], Christine Brun[5,6], Samir Merabet[1,2*]**

[1]Institut de génomique fonctionnelle de Lyon, Centre National de Recherche Scientifique, Lyon, France; [2]Institut de génomique fonctionnelle de Lyon, École normale supérieure de Lyon, Lyon, France; [3]MRC Clinical Sciences Centre, Faculty of Medicine, Imperial College London, Hammersmith Hospital Campus, London, United Kingdom; [4]Institut of Genetics, University of Mainz, Mainz, Germany; [5]Technological Advances for Genomics and clinics, Institut national de la santé et de la recherche médicale, University Aix-Marseille, Parc Scientifique de Luminy, Marseille, France; [6]Centre National de la Recherche Scientifique, Marseille, France

**Abstract** Hox proteins are well-established developmental regulators that coordinate cell fate and morphogenesis throughout embryogenesis. In contrast, our knowledge of their specific molecular modes of action is limited to the interaction with few cofactors. Here, we show that Hox proteins are able to interact with a wide range of transcription factors in the live *Drosophila* embryo. In this context, specificity relies on a versatile usage of conserved short linear motifs (SLiMs), which, surprisingly, often restrains the interaction potential of Hox proteins. This novel buffering activity of SLiMs was observed in different tissues and found in Hox proteins from cnidarian to mouse species. Although these interactions remain to be analysed in the context of endogenous Hox regulatory activities, our observations challenge the traditional role assigned to SLiMs and provide an alternative concept to explain how Hox interactome specificity could be achieved during the embryonic development.

*For correspondence: samir.
merabet@ens-lyon.fr

†These authors contributed
equally to this work

Competing interests: The
authors declare that no
competing interests exist.

Reviewing editor: Diethard
Tautz, Max Planck Institute for
Evolutionary Biology, Germany

## Introduction

There is mounting evidence that many protein–protein interactions (PPIs) are mediated by small peptide motifs called linear motifs (LMs) or eukaryotic/short linear motifs (ELMs/SLiMs) (*Neduva and Russell, 2005*; *Van Roey et al., 2012*, *2014*; *Tompa et al., 2014*). These compact interaction interfaces are typically less than 10 residues in length and are often located within intrinsically disordered regions of highly connected proteins. Due to their small size, SLiMs exhibit high evolutionary plasticity and mediate interactions with many different types of proteins. Moreover, SLiMs are known to be important for the rewiring of interaction networks, being the subject of tissue-specific regulatory mechanisms (*Buljan et al., 2013*).

The contribution of SLiMs to the functional diversification and specification of key regulatory TFs throughout development and evolution remains poorly understood. For example, very few SLiMs listed in the current databases relate to the regulatory interaction between transcription factors (TFs) (*Dinkel et al., 2014*), suggesting that this particular type of functional interaction is more difficult to capture than others. More generally, classic large-scale screening methods based on affinity purification followed by mass spectrometry (AP-MS) or yeast two-hybrid are more efficient for

**eLife digest** In all animals, it is important that cells are correctly organised into tissues and organs. This organisation starts in the embryo, and cells are instructed to perform different roles depending on their position within the body.

A family of proteins called the Hox proteins coordinates the organisation of the cells in the animal embryo by binding to and controlling the expression of specific genes. To properly control their target genes, Hox proteins need to interact with other proteins called transcription factors that can also bind to the genes. However, only a few of these transcription factors have been identified so far, and it is not clear how Hox proteins are able to interact with them.

Here, Baëza, Viala, Heim et al. identified several more transcription factors that can bind to the Hox proteins in fruit fly embryos. The experiments show that Hox proteins are able to bind to many transcription factors that are very different from each other. Baëza, Viala, Heim et al. also show that two short sections within the Hox proteins known as short linear motifs are important for controlling these interactions. A fly Hox protein that was missing these motifs was able to interact with new transcription factors. This inhibitory role was found in Hox proteins from mice and sea anemones, suggesting that these motifs may play the same role in all animals.

Baëza, Viala, Heim et al.'s findings challenge the traditional view of the role of the short linear motifs in interactions between proteins. Also, the findings provide an alternative explanation for how the Hox proteins are only able to interact with particular transcription factors in animal embryos. The next step will be to find out whether the inhibitory role of short linear motifs could more generally apply to many other protein families.

detecting stable interactions between structured domains than for revealing transient interactions involving SLiMs (*Landry et al., 2013*). Therefore, alternative approaches are needed to decipher SLiM-mediated interactions and functions within the context of developmental regulatory networks in vivo.

Here, we tackle this issue by using Hox proteins as a case study. Hox proteins are homeodomain (HD)-containing TFs present in all cnidarian and bilaterian species (*Finnerty, 2003*). They are required throughout the embryogenesis for controlling specific cell fates and structures along different axes and in territories as different as the limb bud (*Zakany and Duboule, 2007*), cardiac outflow tract (*Bertrand et al., 2011*), and female genital disc (*Foronda et al., 2005*). The specific functions of Hox proteins in vivo contrast with their ability to recognize closely similar DNA-binding sites as monomers in vitro (*Berger et al., 2008*; *Noyes et al., 2008*). This so-called Hox paradox strongly suggests that additional cofactors are required for helping Hox proteins to elicit their diverse and specific transcriptional programs in vivo.

To date, only one type of cofactors is described to specify Hox functions at the molecular level. These cofactors are collectively referred to as the PBC class of HD-containing TFs, and correspond to the Pbx1-4 and Extradenticle (Exd) proteins in mammals and *Drosophila melanogaster*, respectively (*Mukherjee and Bürglin, 2007*). Biochemical studies have shown that the interaction between Hox and PBC proteins relies on a highly conserved motif of Hox proteins, the hexapeptide (HX) (*Mann et al., 2009*). The HX motif constitutes the generic signature of Hox proteins after key positions within the HD (*Merabet et al., 2009*). It belongs to the LIG type of SLiM according to the ELM database (http://elm.eu.org/) and contains a core YF/PWM sequence conserved in all but posterior Hox paralogs throughout animal evolution (*Merabet et al., 2009*). More generally, the HX motif has been defined as corresponding to an invariant Tryptophan residue located in a hydrophobic environment and followed by basic residues from +2 to +5 (*In der Rieden et al., 2004*). Crystal structures with truncated proteins emphasized the importance of the Tryptophan residue in establishing strong interactions with specific residues of the PBC partner (*Mann et al., 2009*). However, in vivo analyses showed that mutations within the HX motif (including that of the key Tryptophan residue) did not systematically abolish PBC-dependent functions of Hox proteins (*Galant et al., 2002*; *Merabet et al., 2003*). In addition, Hox-PBC interactions are influenced by the promoter environment and can occur in absence of the HX motif in several cnidarian and bilaterian Hox proteins (*Hudry et al., 2012*, *2014*). Reciprocally, the HX is also required for PBC-independent functions (*Merabet et al., 2011*) and for

interacting with Bip2, a TATA-binding protein associated factor in *Drosophila* (*Prince et al., 2008*). Together these observations highlight that the HX motif is neither a unique nor an obligatory Hox protein interface for recruiting the PBC cofactor, suggesting that Hox-PBC interactions could rely on the presence of other specific SLiM(s). Notably, another motif called UbdA and present in Ultrabithorax (Ubx) and AbdominalA (AbdA) proteins of protostome lineages was recently described to be important for the formation and activity of the Ubx/Exd complex in *Drosophila* (*Merabet et al., 2007*; *Hudry et al., 2012*; *Foos et al., 2015*).

In summary, our current knowledge on SLiM-mediated interactions in Hox proteins is limited to only two different types of TFs, the PBC and Bip2 proteins. Given the number of embryonic events controlled by Hox proteins, we hypothesize that Hox SLiMs such as the HX and UbdA motifs could interact with a higher number of TFs. Identifying these TFs represents a major challenge to understand part of the molecular cues underlying Hox transcriptional specificity and diversity in vivo.

Here, we exploited the recently developed bimolecular fluorescence complementation (BiFC) (*Hudry et al., 2011*) to profile a wide range of Hox protein interactions in the *Drosophila* embryo and investigate whether SLiMs could influence their specificity in vivo. As a first step, we identified the respective sets of BiFC interactors of five *Drosophila* Hox paralogs, showing that each Hox interactome relies on a different combination of TFs. The role of the HX and UbdA motifs was then analysed in several Hox interactomes and in different tissues of the live embryo. Our data establish that the ablation of Hox SLiMs not only prevents several interactions but additionally leads to a number of ectopic interactions. These effects differ depending on the Hox protein and tissue considered, suggesting that SLiM activity could be strongly influenced by the protein environment. Furthermore, results obtained with mouse and cnidarian Hox proteins indicate that the inhibitory activity of SLiMs could be important for restricting the inherent binding potential of intrinsically disordered regions.

Altogether, these findings provide new insights on how Hox transcriptional specificity could be reached in vivo and add to the functional repertoire of SLiMs.

## Results

### A competitive BiFC screen reveals new candidate binding partners of the Hox protein AbdA in the live *Drosophila* embryo

BiFC relies on the property of monomeric fluorescent proteins to be reconstituted from two separate sub-fragments upon spatial rearrangement (*Ghosh et al., 2000*). This property is used with different types of proteins in various cell and animal model systems to demonstrate the close proximity hence the existence of possible interactions between two putative protein partners (*Kerppola, 2008*; *Kodama and Hu, 2012*).

We previously demonstrated that BiFC was sensitive and specific enough for analysing Hox-TF interactions in the live *Drosophila* embryo (*Hudry et al., 2011*). Experimental parameters were established by using the partnership between AbdA and Exd as a case study. Interaction was visualized by fusing the two partners with complementary fragments of the Venus (yellow: *Figure 1A–A′*), mCherry (red) or Cerulean (blue) fluorescent protein. Among several controls, we showed that the simultaneous co-expression of a 'cold' AbdA protein (i.e., not fused with a fragment of the fluorescent protein) with AbdA and Exd fusion proteins could induce a titration of the BiFC complex (*Hudry et al., 2011*). Thus, cold interactions (in this case AbdA–AbdA and AbdA–Exd interactions) could compete against BiFC, leading to a significant decrease of fluorescent signals in the embryo (*Hudry et al., 2011*).

We reasoned that any protein capable of displacing the AbdA and/or Exd fusion protein from the BiFC complex could lead to a loss of the fluorescence. This readout could thus serve to rapidly identify putative interacting partners of AbdA. In this case, competitive BiFC could not be observed with proteins that are exclusively participating in the AbdA/Exd complex (*Figure 1B–B′*). In addition, the titration of BiFC signals could only occur when the cold interaction is strong enough to disrupt the assembly between AbdA and Exd (*Figure 1C–C′*). However, as competitive BiFC could not discriminate between AbdA- and Exd-specific interacting partners (*Figure 1C–C′*), a second experimental phase will be necessary to confirm the Hox interaction status. Despite these limitations, we decided to test our hypothesis with a reasonable number of candidate TFs and by using a fast genetic approach. To this end, we established a BiFC reporter fly line expressing AbdA and Exd fusion

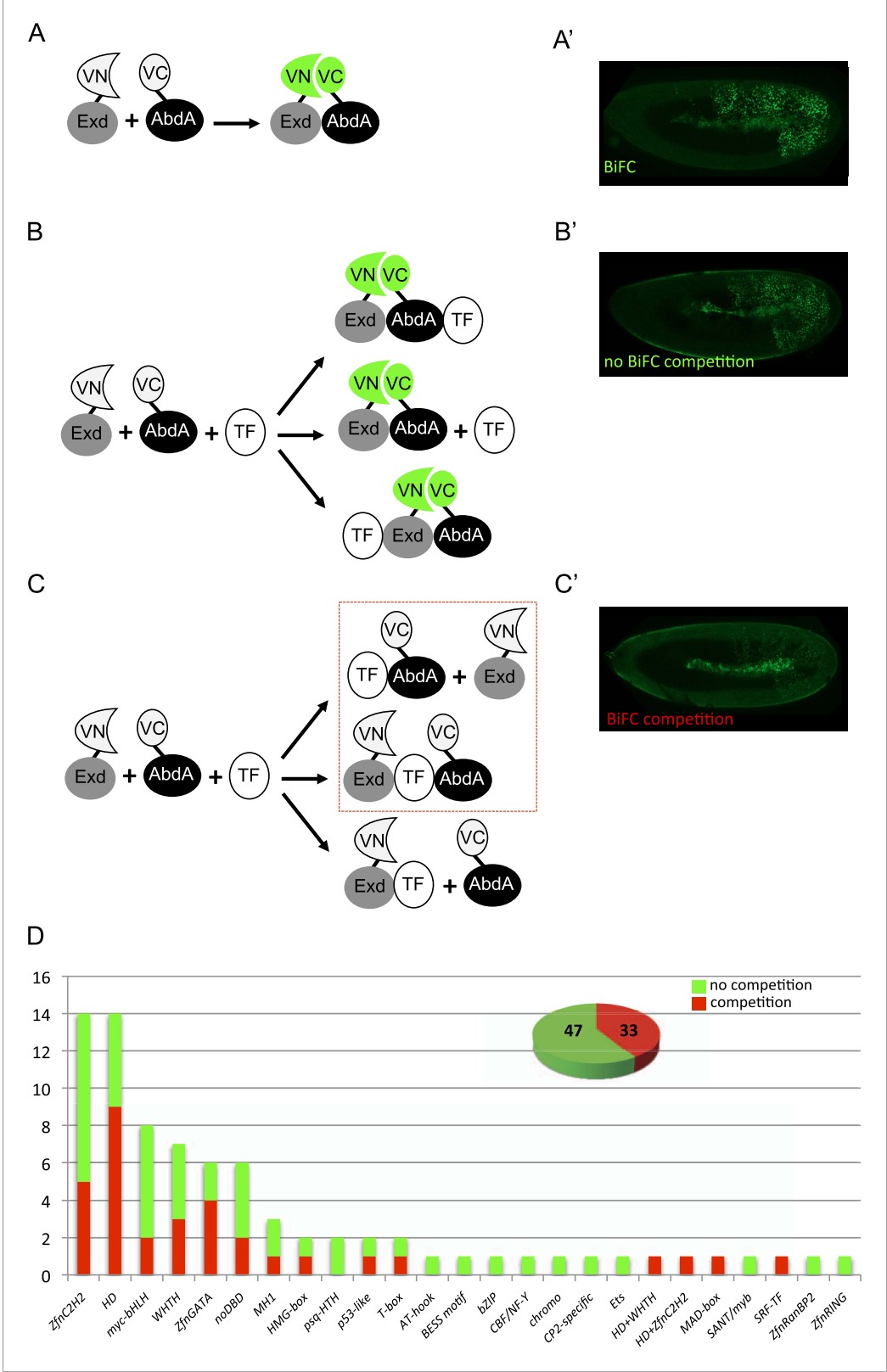

**Figure 1**. A BiFC competition screen identifies candidate transcription factors (TFs) as potential binding partners of the Hox protein AbdominalA (AbdA). (**A–C**) Principle of the competition test. (**A–A'**) Co-expression of Extradenticle (Exd) and AbdA proteins fused to the N-(VN) or C-(VC) terminal fragment of the Venus fluorescent protein leads to

*Figure 1. continued on next page*

*Figure 1. Continued*

BiFC in the embryo. (**B–B'**) Cases where no BiFC competition could be observed with a cold TF. **B'** is an illustrative picture of non-competitive BiFC resulting from the simultaneous co-expression of the red fluorescent protein RFP (see also *Figure 1—figure supplement 2*). (**C–C'**) Cases where BiFC competition could be observed with a cold TF. **C'** is an illustrative picture of competitive BiFC resulting from the simultaneous co-expression of a nuclear-localized form of Exd (see also *Figure 1—figure supplement 1*). Note that AbdA-interacting partners do not obligatory lead to competitive BiFC. TFs that could be validated in the secondary step as AbdA-binding partners (see *Figure 2*) are indicated (dotted-red box). (**D**) Graph showing the repartition of competitive (red bars) and non-competitive (green bars) TFs with regard to their DNA-binding domain. See also *Supplementary file 1*.

The following figure supplements are available for figure 1:

**Figure supplement 1**. Illustrative pictures of competitive TFs.

**Figure supplement 2**. Illustrative pictures of non-competitive TFs.

proteins under the control of the *abdA-Gal4* driver ('Materials and methods' and [*Hudry et al., 2011*]). This fly line can be crossed with individuals containing any UAS-driven cold candidate-binding partner and BiFC could directly be assessed in the embryo progeny. As a control, we verified that the co-expression of a nuclear-localized form of Exd (*Kammermeier et al., 2004*) could indeed affect BiFC fluorescent signals in the embryo (*Figure 1—figure supplement 1*). In comparison, co-expressing a red fluorescent protein (RFP) under the same condition did not lead to any changes in the BiFC profile (*Figure 1—figure supplement 2*).

Here, we deliberately focused the screen on TFs that could participate in the transcriptional programs of Hox proteins in their various developmental contexts. We chose a starting set of 80 TFs covering different DNA-binding families and displaying distinct expression profiles in the three main germ layers of the embryo (*Supplementary file 1*). We observed that 33 of those TFs could compete against AbdA/Exd assembly (*Figure 1D* and *Figure 1—figure supplements 1, 2*). Among them we found Biniou (Bin) and Mad, previously described to participate in the regulation of Hox target enhancers (*Grienenberger et al., 2003*; *Walsh and Carroll, 2007*), and Teashirt (Tsh), known to help Hox proteins in specifying the trunk segments of the embryo (*Fasano et al., 1991*). Thus, these three TFs validate the competition screen. Interestingly, 65% of the tested HD and GATA TFs (9/14 and 4/6, respectively) were positive in the competition test (*Figure 1D*), representing a strong tendency compared to the other tested TF classes. In total, none of the 33 competitors was previously described as a binding partner of AbdA or Exd, illustrating the efficiency of the competition screen for revealing new candidate cofactors in vivo.

## The majority of AbdA-TF interactions can occur in different cell-contexts

Our genetic competitive approach revealed proteins that could potentially bind to AbdA and/or Exd. We next analysed whether these positive competitors could more specifically interact with AbdA in a complementary BiFC-based approach. To this end, we generated fly lines carrying each corresponding TF as a UAS-driven fusion construct compatible for BiFC (*Figure 2A* and 'Materials and methods'). Two additional TFs (TFIIbeta and Knot [Kn]) were added to the 33 positive competitors, reaching a total of 35 fusion TFs that could be used for BiFC in *Drosophila* (*Figure 2B* and *Supplementary file 1*). TFIIbeta, could not be tested by competition, as no corresponding UAS-driven fly line exists. It is however a good positive candidate as it is described to interact with Ubx in a yeast two-hybrid screen (*Bondos et al., 2004*). Kn did not compete against BiFC in the first step and was used as a negative candidate interacting partner of the experiment.

BiFC was observed in the epidermis of stage 10 embryos, even for TFs that are not endogenously expressed in this tissue (see 'Materials and methods', *Supplementary file 2* and [*Hammonds et al., 2013*]). We anticipated that the epidermis was appropriate for the interaction since competition was observed in this tissue. In addition, the epidermis has been shown to tolerate the activation of mesodermal target genes upon the ectopic expression of mesoderm-specific TFs (*Cunha et al., 2010*), suggesting it is a relatively neutral tissue. Finally, BiFC could also be increased in a heterologous tissue because of the absence of competition by the endogenous gene product, as previously described (*Hudry et al., 2011*).

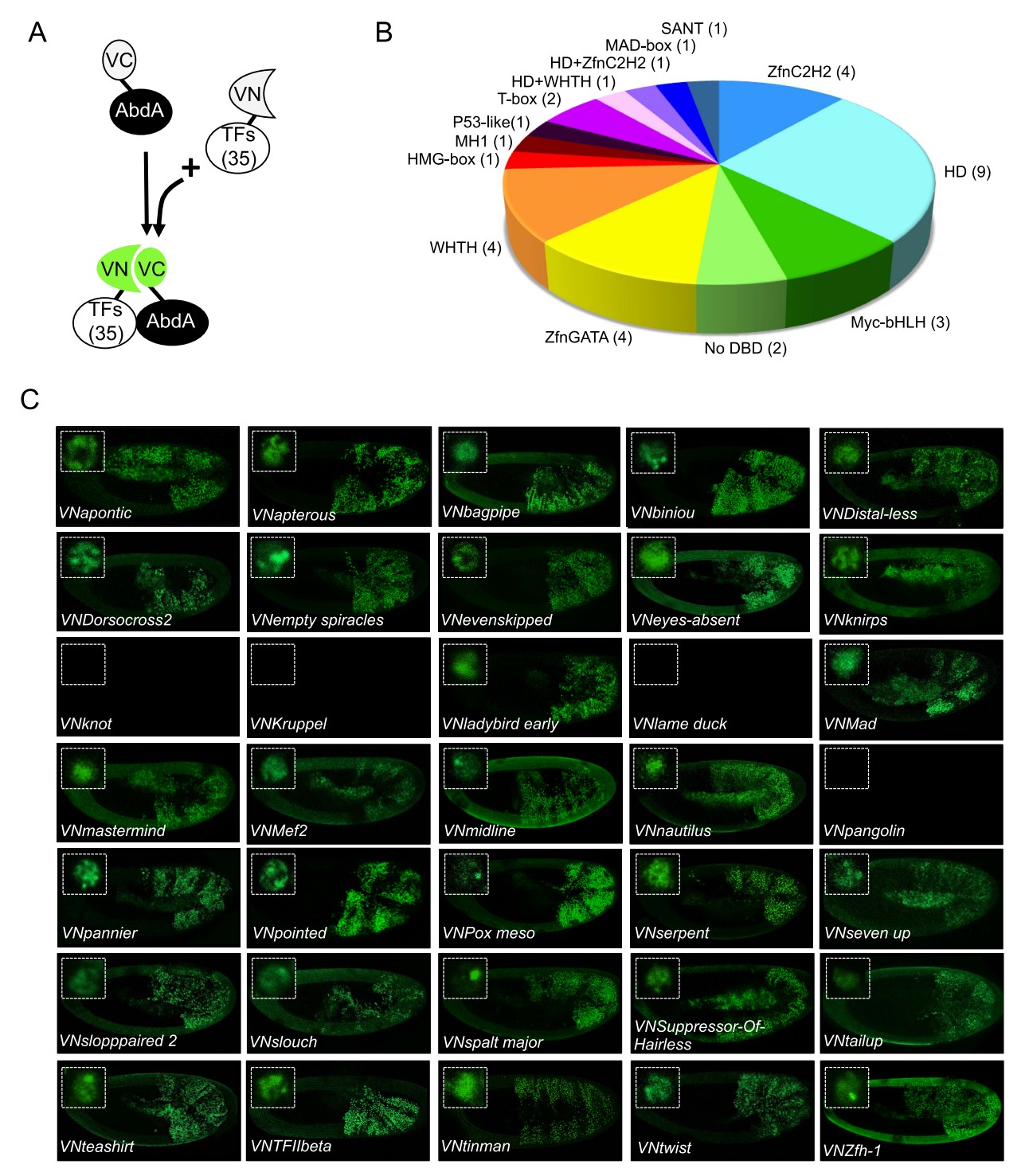

**Figure 2**. BiFC validates the AbdA-interaction status of competitive TFs. (**A**) Principle of the BiFC screen between AbdA and the 35 selected TFs. (**B**) Repartition of the 35 selected TFs with regard to their DNA-binding domain. (**C**) Illustrative pictures of BiFC signals obtained between VC-AbdA and the indicated VN-TF in the epidermis of stage 10–12 of live embryos. Fusion constructs are expressed with the *abdA-Gal4* driver. Note that typical nuclear interaction profiles are observed between AbdA and different TFs (white-dotted boxes). See also 'Materials and methods', *Supplementary files 2, 3* and *Figure 2—figure supplements 1, 2*.

The following figure supplements are available for figure 2:

**Figure supplement 1**. Co-immunoprecipitation (co-ip) between AbdA and TFs selected from the set used for BiFC in the *Drosophila* embryo.

*Figure 2. continued on next page*

*Figure 2. Continued*

**Figure supplement 2**. BiFC between mesodermal TFs and AbdA in the mesoderm.

Among the 35 TFs tested as fusion constructs, 31 led to BiFC signals with AbdA, including TFIIbeta (*Figure 2C*). These fluorescent signals display homogenous or punctuate distribution within the nucleus, depending on the TF considered (*Figure 2C*). BiFC was negative with Kn as expected, given that no competition was previously observed with this TF. BiFC was however also negative with Krüppel (Kr), Lameduck (Lmd), and Pangolin (Pan), although these three TFs were positive competitors. This discrepancy suggests that the previously observed competition could result from the formation of specific cold complexes with the Exd and not with the AbdA fusion protein. Alternatively, the fusion topologies could forbid the interaction hence BiFC between the three TFs and AbdA. Indeed, the negative influence of fusion topologies on protein–protein interactions was previously described and is hardly predictable (*Hudry et al., 2011*).

To assess the potential influence of fusion topologies on AbdA-TF interactions, we performed co-immunoprecipitation (co-ip) experiments, using an anti-HA antibody recognising a HA-tagged form of AbdA. We reasoned that a small HA epitope (8 residues long) should be more neutral than the Venus fragment (80 residues long) for the interaction with the TF. Practically, the fusion TF was co-expressed with the AbdA-HA construct in S2 cells, and its presence was verified with an anti-GFP antibody recognizing the Venus fragment (see 'Materials and methods'). Experimental parameters were established with the control Exd cofactor. All tested BiFC-positive TFs were found by co-ip (*Supplementary file 3*), highlighting that the S2 cell environment is appropriate for revealing interactions with tissue-specific TFs. Thus, observations from BiFC could be reproduced by co-ip, as previously noticed (*Lee et al., 2011*). Co-ip was also performed with the three positive competitors that did not produce BiFC with AbdA (Kr, Lmd, Pan). We observed that these three TFs could be immuno-precipitated with AbdA (*Supplementary file 3*). This was not the case for Kn, which was negative both in the competition and BiFC tests (*Figure 2—figure supplement 1*). We conclude that inappropriate fusion topologies are likely to be responsible for the absence of BiFC between AbdA and Lmd, Kr or Pan in the epidermis.

Several of the tested TFs are not expressed in the epidermis (*Supplementary file 2*). We thus wondered whether the interaction with those TFs could also be reproduced in their endogenous expression tissue. To this end, we repeated BiFC analyses in the mesoderm, using mesodermal TFs that are not expressed in the epidermis, including Lmd and Kn (*Supplementary file 3*). We observed that TFs interacting with AbdA in the epidermis were also positive in the mesoderm (*Figure 2—figure supplement 2*). Fluorescent signals were however generally weaker than in the epidermis, probably due to the competition by the endogenous gene products. Surprisingly, weak BiFC signals could also be observed with Kn, while Lmd remained negative in the mesoderm (*Figure 2—figure supplement 2*). We concluded that Kn could interact with AbdA but that this interaction is more sensitive to the cell environment in order to occur.

In summary, although our approach was performed with a limited set of TFs (around 12% of all *Drosophila* TFs), it revealed an unexpectedly high number of new binding partners of AbdA. However, whether and how these binding partners could be used in the context of endogenous regulatory activities of AbdA remains to be investigated. This result still illustrates the strong propensity of AbdA to establish interactions with diverse TFs in vivo. In the following, we considered the set of 35 TFs as sufficiently representative for addressing the issue of the molecular mechanisms underlying Hox interactome specificity in vivo.

## Regions outside the homeodomain influence dependent and independent DNA-binding interactions of AbdA with TFs

Our experimental parameters allow quantifying subtle changes in fluorescent signal intensities in the whole *Drosophila* embryo ('Materials and methods' and [*Hudry et al., 2011*]). These variations in the fluorescence intensity can be correlated to differences in interaction affinity. Indeed, high affinity partners lead to fast accumulation, hence strong BiFC signals, and vice versa (*Hudry et al., 2011*). Here, levels of BiFC were used to measure the effects of AbdA mutations on the interaction potential

with each of the 35 TFs. We first focused on the HD, which is responsible for the DNA-binding and the most conserved part of Hox proteins. The HD of Hox proteins is also described to interact with different types of cofactors (*Merabet et al., 2009*), suggesting it could be involved in several of the observed interactions. More precisely, we asked (i) whether AbdA-TF interactions could depend on the DNA-binding activity of the HD, and (ii) whether the HD could be sufficient for AbdA-TF interactions. BiFC was performed with two corresponding mutant forms of AbdA: one carrying the N51A mutation in the HD, which abolishes the DNA-binding activity of full length AbdA (AbdA51) (*Hudry et al., 2011*), the other resulting in a truncated version that contains only the HD (AbdAHD) (*Boube et al., 2014*). The two fusion constructs were inserted on the same genomic locus and expressed at similar levels in the embryo as the wild type AbdA fusion protein ('Materials and methods' and [*Hudry et al., 2011*; *Boube et al., 2014*]). BiFC was measured in the epidermis and considered as affected when the fluorescent signal was equal or lower than 50% of the fluorescent level normally obtained with wild type AbdA (see 'Materials and methods' for quantification details).

Among the 31 BiFC-positive interactions, 18 were strongly affected or lost with AbdA51 (compared *Figure 3A,B*, and *Figure 3—figure supplement 1*). Still, a significant proportion of the interactions (for 13 TFs) was retained, two of which were even stronger than with wild type AbdA (leading to intensities higher than 120% of the wild type fluorescent signal). Thus, the two corresponding TFs (Pannier and Slouch) preferentially interact with a form of AbdA that is unable to bind DNA. Effects were more drastic with the minimal AbdAHD construct, which kept only eight interactions among the 31 positive TFs (compare *Figure 3A,C*, and *Figure 3—figure supplement 2*). One of these interactions (with Serpent, Srp) was also stronger compared to wild type AbdA.

The observation of stronger interactions in some cases prompted us to analyse whether previously negative TFs with AbdA (Kn, Kr, Lmd, and Pan) could produce BiFC with either of the two AbdA mutant forms. We found that Kr could indeed lead to BiFC signals when using the AbdAHD version (*Figure 3C* and *Figure 3—figure supplement 2*). Thus, the interaction was strong enough in this particular case to be visualized despite unfavourable fusion topologies, illustrating that HD-surrounding region(s) could inhibit the interaction potential of the HD in vivo.

Together these results show that the DNA-binding of AbdA is not systematically required for recruiting the TFs. The HD itself is also not sufficient in most cases, suggesting that surrounding protein region(s) are important for recruiting TFs. Interestingly, these HD-surrounding region(s) can also have an inhibitory role since their absence allows the formation of stronger interactions between the HD and two of the tested TFs. Given that HD-surrounding regions are less conserved in Hox proteins in general, we then asked whether some of the revealed interactions could also be found with other *Drosophila* Hox proteins.

## Specific combinations of TFs underlie the formation of different Hox interactomes in vivo

With the exception of Ubx, *Drosophila* Hox proteins have few redundant functions with AbdA, as reflected at the protein sequence and embryonic expression levels (*Figure 4A*). We thus wondered whether common vs specific features between Hox proteins could be found in their respective interactomes. To this end, we repeated BiFC between the 35 TFs and four other Hox proteins (see 'Materials and methods'), namely Sex combs reduced (Scr), Antennapedia (Antp), Ultrabithorax (Ubx), and AbdominalB (AbdB).

Overall, we observed an unexpected high proportion of positive interactions with the four additional Hox proteins: from 22 and 18 with Scr and Antp, to 21 and 26 with Ubx and AbdB, respectively (*Figure 4B* and *Figure 4—figure supplements 5–9*, *Figure 5—figure supplement 1*). Because of this high proportion, the majority of interactions are common to several Hox proteins. Despite this number of common interactions, each Hox interactome contains a specific combination of binding partners. Interestingly, the hierarchical clustering of Hox interactomes (see 'Materials and methods') does not reflect the protein sequence similarity between Hox proteins (*Figure 4A–B*). For example, the interactomes of AbdA and AbdB appear closely similar although AbdB is much more divergent from AbdA than the other Hox proteins.

Taken together these observations show that Hox proteins are able to bind to common and different types of TFs. Still, each Hox interactome contains a distinct combination of TFs, demonstrating a certain

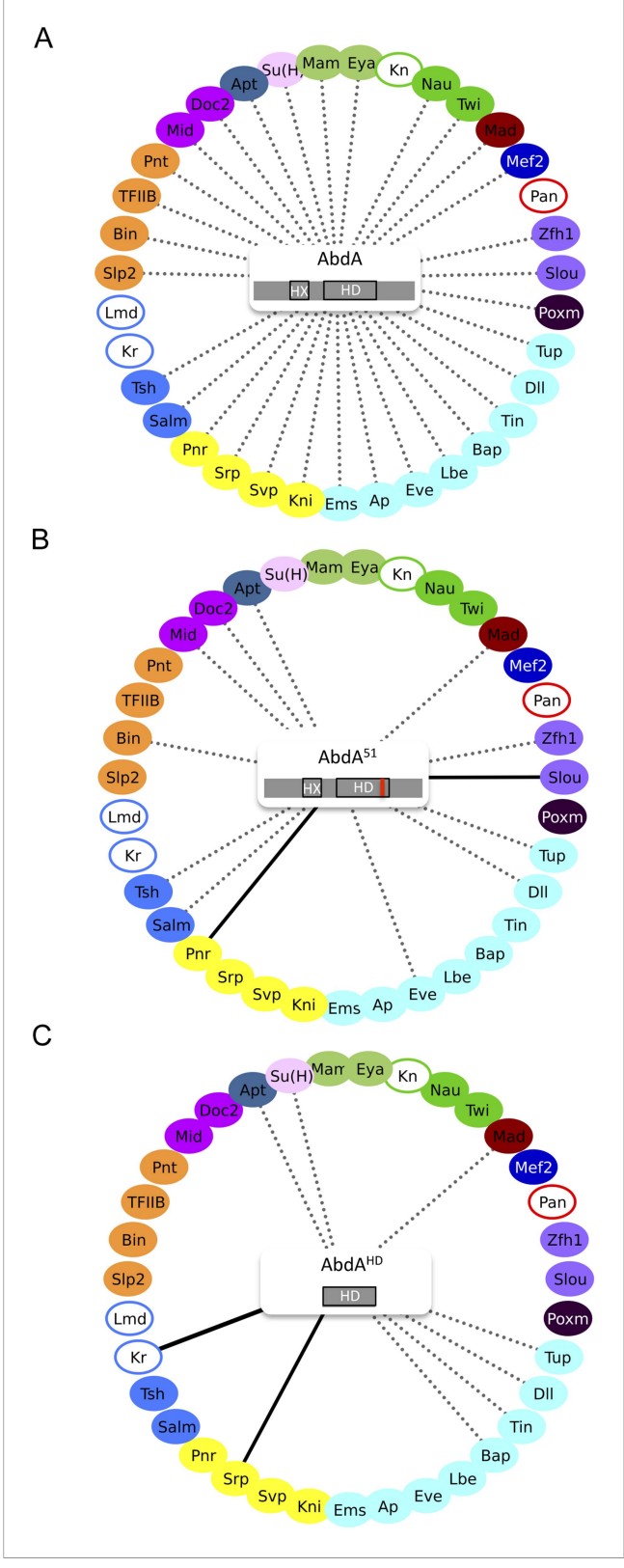

**Figure 3**. Role of the homeodomain (HD) in the AbdA interactome. (**A**) Interactome with wild type AbdA. (**B**) Interactome with the DNA-binding deficient form of AbdA (mutated in the residue 51 of the HD, as illustrated with the red bar). See also *Figure 3—figure supplement 1*. (**C**) Interactome with the HD of AbdA. See also *Figure 3—figure supplement 2*. Each interactome is represented with the 35 TFs. The colour code for TFs

*Figure 3. continued on next page*

*Figure 3. Continued*

corresponds to their type of DNA-binding domain, as shown in the *Figure 2B*. TFs that are not colour-filled correspond to TFs that do not interact with the wild type Hox protein. Those TFs are not connected to the Hox protein. Dotted lines indicate TFs that do interact with the wild type Hox protein. Interactions with Hox variants are depicted as the following: dotted lines indicate unaffected interactions (in between 51% and 119% of the wild type interaction); solid black lines indicate stronger (equal or superior to 120% of the wild type interaction) or novel interactions (with a non-colour-filled TF); absence of the dotted line with a colour-filled TF indicates a partial (equal or below to 50% of the wild type interaction) or complete loss of the interaction. Each Hox variant is schematized in the centre of the interactome. HD: Homeodomain. HX: Hexapeptide. See also 'Materials and methods'.

The following figure supplements are available for figure 3:

**Figure supplement 1**. BiFC between the 35 TFs and the homeodomain (HD)-mutated form of AbdA.

**Figure supplement 2**. BiFC between the 35 TFs and the homeodomain (HD) of AbdA.

---

level of specificity. Moreover, the hierarchical clustering of Hox interactomes suggests that the recruitment of common cofactors would not obligatorily rely on the same Hox protein interface.

## The common Hexapeptide (HX) motif is differently used within each Hox interactome in vivo

BiFC analyses with AbdA showed that HD-surrounding regions are important for most of the revealed interactions. In addition to the HD, the HX motif represents the second generic signature of Hox proteins. We thus asked whether this common Hox SLiM could be important for recruiting common TFs in vivo. Its role was assessed within each Hox interactome by repeating BiFC between the 35 TFs and the corresponding HX-mutated Hox proteins (see 'Materials and methods' and [*Hudry et al., 2012*]).

Heatmap representation shows that the HX mutation led to a complete reorganisation of Hox interactomes when compared to the wild type Hox proteins (compare *Figure 4B,C* and *Figure 4—figure supplements 5–9*). More precisely, the HX mutation affects the majority of Hox-TF interactions. Surprisingly, this mutation leads not only to a loss, but also to a gain of the Hox interaction potential, with the appearance of stronger or new interactions (highlighted in yellow in *Figure 4C*). The balance between gain and loss was different depending on the Hox protein: Antp and AbdB were more sensitive to a loss while the reverse was observed with Ubx and AbdA. The HX of Scr was equally responsible for a gain or loss of interactions (*Figure 5* and *Figure 5—figure supplement 1*). Overall, the HX mutation leads more often to a gain than a loss of the Hox interaction potential. Interestingly, the positive or negative influence of the HX on Hox-TF interactions is not identical for each TF. For example, the HX mutation has no effect on AbdA-Twist (Twi) interaction, while it leads to a loss of interaction between Antp and AbdB and the same TF, and to a stronger and novel interaction with Ubx and Scr, respectively (*Figure 5* and *Figure 5—figure supplement 1*). Thus, the role of the HX appears dictated by the Hox protein to which it belongs and not by the interacting TF, therefore reinforcing the fact that HX neighbourhood is important for controlling its interaction properties (*Merabet and Hudry, 2011*).

In summary, our results show that a short conserved motif, common to all Hox proteins, is specifically used both for promoting and limiting their interaction potential with TFs.

## Common vs paralog-specific Hox protein motifs display different interaction properties with TFs

Since we observed that the HX could inhibit the interaction potential of Hox proteins, we asked whether this property could also be found with other more specific Hox SLiMs. We focused on Ubx and AbdA, in which the HX mutation led to the highest number of gained interactions among the tested Hox proteins. Interestingly, Ubx and AbdA share the UbdA motif, which is conserved in most protostome lineages (*Balavoine et al., 2002*). This motif was shown to be important for recruiting the Exd cofactor in Ubx (*Merabet et al., 2007*; *Foos et al., 2015*) and for tissue-specific activities of AbdA in vivo (*Merabet et al., 2011*).

As previously done with the HX motif, the role of the UbdA motif was assessed by performing BiFC with UbdA-mutated forms of Ubx and AbdA (*Figure 6A* and [*Hudry et al., 2012*]). We focused on the

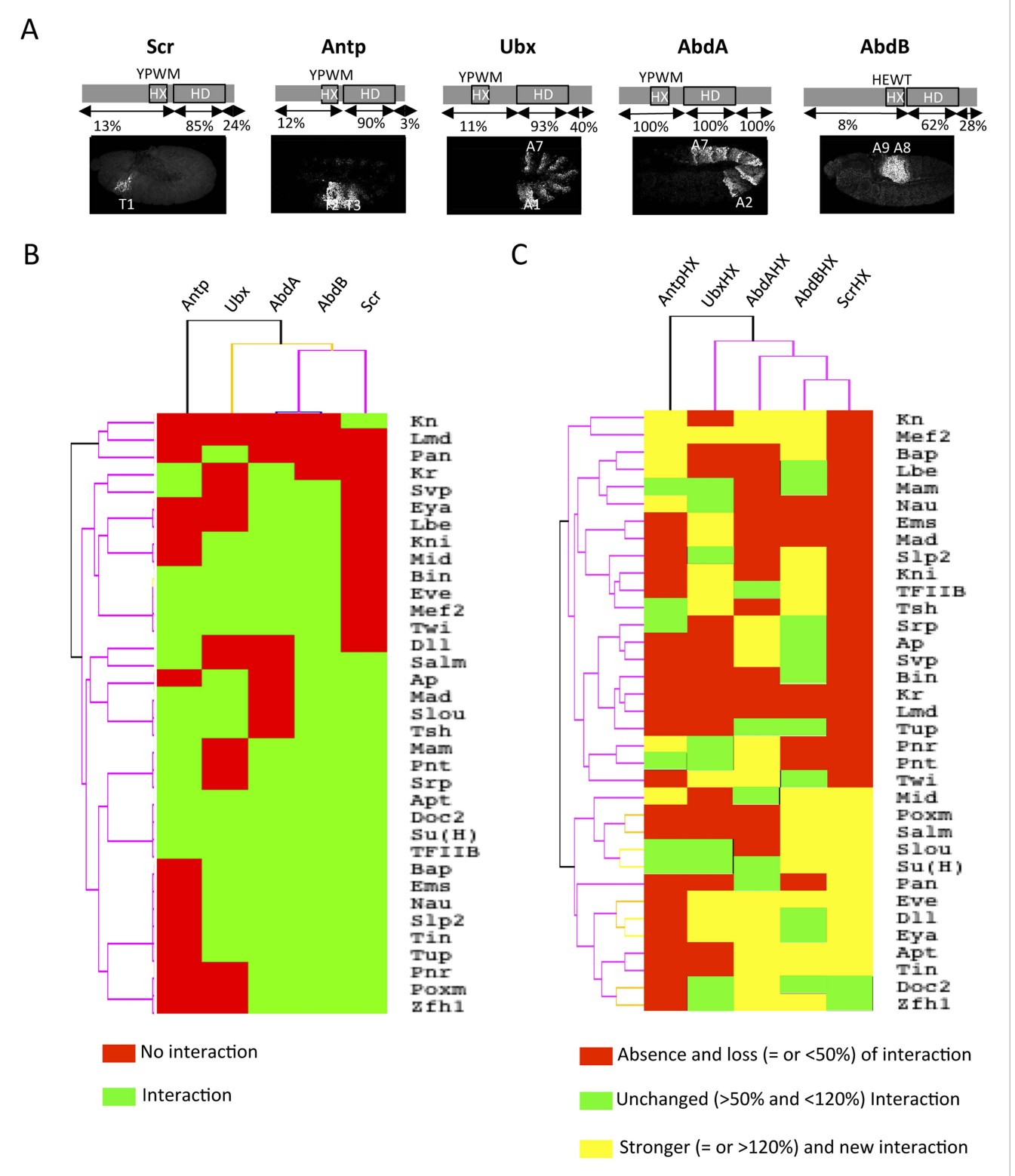

**Figure 4**. Comparison between wild type and Hexapeptide(HX)-mutated Hox interactomes. (**A**) Embryonic expression profile (grey) and protein sequence identity of each of the five *Drosophila* Hox proteins under study. The percentage of sequence identity is given in comparison to AbdA. The conserved core sequence of the HX is also given for each Hox protein. (**B**) Heatmap showing the organisation of wild type Hox interactomes with the 35 TFs. See also *Figure 4—figure supplements 1–4*. (**C**) Heatmap showing the organisation of HX-mutated Hox interactomes with the 35 TFs. See also *Figure 4—figure supplements 5–9*. Interactions are symbolized by a colour code, as indicated. Note that the yellow colour, which corresponds to a gain of the interaction

*Figure 4. continued on next page*

Figure 4. Continued

potential, appears with the HX mutation in all Hox proteins. Dendogram branches are coloured according to their bootstrap score: black 100%, grey 90–100%, blue 80–90%, green 70–80%, yellow 60–70%, orange 50–60%, pink 0.1–50%, red 0% support respectively.

The following figure supplements are available for figure 4:

**Figure supplement 1**. BiFC between the 35 TFs and Sex combs reduced (Scr).

**Figure supplement 2**. BiFC between the 35 TFs and Antennapedia (Antp).

**Figure supplement 3**. BiFC between the 35 TFs and Ultrabithorax (Ubx).

**Figure supplement 4**. BiFC between the 35 TFs and AbdominalB (AbdB).

**Figure supplement 5**. BiFC between the 35 TFs and hexapeptide (HX)-mutated Scr.

**Figure supplement 6**. BiFC between the 35 TFs and hexapeptide (HX)-mutated Antp.

**Figure supplement 7**. BiFC between the 35 TFs and hexapeptide (HX)-mutated Ubx.

**Figure supplement 8**. BiFC between the 35 TFs and hexapeptide (HX)-mutated AbdA.

**Figure supplement 9**. BiFC between the 35 TFs and hexapeptide (HX)-mutated AbdB.

20 common binding partners of Ubx and AbdA to potentially reveal a common usage mode of the UbdA motif between the two Hox proteins.

Results show that the UbdA mutation affects the majority of interactions in both Hox proteins, as previously noticed for the HX mutation (*Figure 6B–C* and *Figure 6—figure supplements 1, 2*). Effects can also be categorized as a gain or a loss of the interaction potential. However, the UbdA motif has a more pronounced tendency to be required for the interaction rather than for inhibiting it (*Figure 6B–C*). For example, the UbdA mutation leads to 12 loses and 5 gains among the 20 tested TFs with AbdA. In comparison, 3 losses and 12 gains were induced upon the HX mutation for the same set of interactions. In addition, the HX and UbdA mutations have distinct effects for the majority of interactions established by Ubx or AbdA (*Figure 6B–C*). This result highlights that the two Hox proteins do not use the HX and UbdA motifs similarly. Interestingly, the UbdA mutation often leads to similar effects in Ubx and AbdA compared to the HX mutation (*Figure 6D–E*). More precisely, the UbdA mutation affects the majority (13/20) of the tested TFs in a similar way, which is not the case for the HX mutation (8/20).

In conclusion, the UbdA motif displays preferential Ubx/AbdA-specific interaction properties compared to the HX motif. These interaction properties rely in part on inhibitory activities, highlighting that the negative influence on PPIs is not a specific property of the HX motif. To gain further insights into the molecular property of Hox SLiMs, we next examined whether the regulatory activity of the HX and UbdA motifs could change depending on the embryonic tissue considered.

## The HX and UbdA motifs have tissue-specific interaction properties

Previous work showed that the HX and UbdA motifs have tissue-specific functions in AbdA (*Merabet et al., 2011*), suggesting that their interaction properties with TFs would not be identical in different embryonic tissues. To test this hypothesis, we analysed the interaction potential of HX- or UbdA-mutated AbdA in the mesoderm and nervous system, focusing on TFs that are normally expressed in one and/or both tissues. We also used a set of TFs that were all BiFC-positive with AbdA in the epidermis and all but one sensitive to the HX or UbdA mutation in this tissue (*Figure 7A*).

We observed that the HX and UbdA motifs were less often required in the mesoderm since their mutation affected fewer interactions than in the epidermis (compare *Figure 7A,B*, and *Figure 7—figure supplement 1*). Still, affected TFs again correspond to a loss or a gain of the Hox interaction potential. The inhibitory activity of the HX and UbdA motifs was more pronounced in the

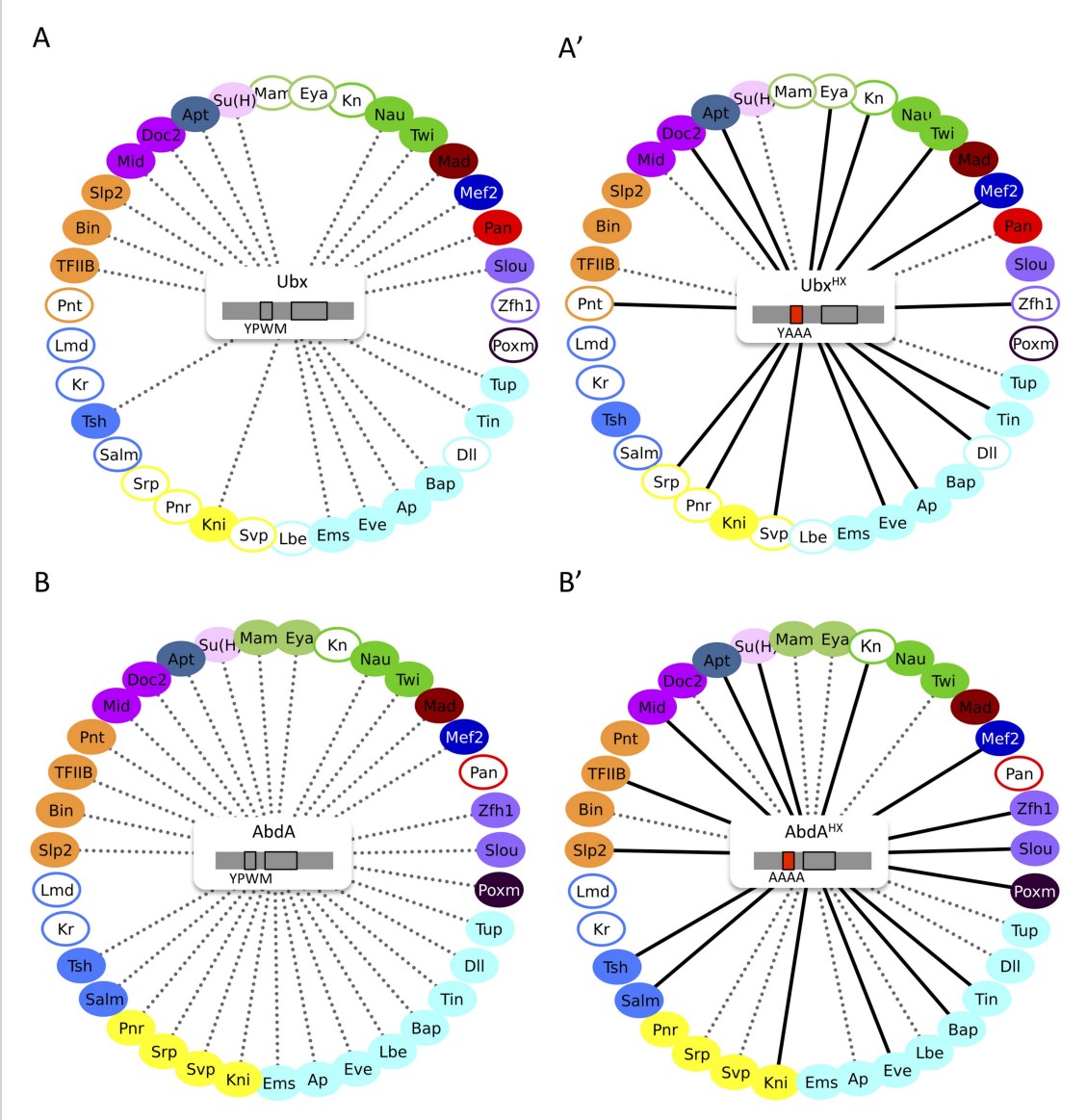

**Figure 5**. The HX mutation increases the interaction potential of Hox proteins with TFs in vivo: example in Ubx and AbdA. (**A–A'**) Comparison between wild type and HX-mutated interactomes of Ubx. (**B–B'**) Comparison between wild type and HX-mutated interactomes of AbdA. The HX mutation led more frequently to stronger or new interactions than to interaction loses in these two Hox proteins. Colour code and representation are as in **Figure 3**. The HX mutation is indicated and highlighted in red. See also **Figure 5—figure supplement 1**.

The following figure supplement is available for figure 5:

**Figure supplement 1**. The HX mutation increases the interaction potential of Hox proteins with TFs in vivo: example in Scr, Antp, and AbdB.

nervous system, since their mutation led only to stronger interactions with TFs (**Figure 7C** and **Figure 7—figure supplement 2**). This effect was more obvious with the HX mutation, which led to a gain of interaction for all but one TF in this tissue (**Figure 7C**).

In total, all tested TFs were not similarly affected by the HX or UbdA mutation in the three different tissues. Thus, the two motifs are differently used depending on the Hox protein and tissue considered. This specific usage mode is based both on the positive and negative control of PPIs. Given the evolutionary conserved roles of Hox proteins in general, we then tested whether these novel facets of SLiM activity could also be found in Hox proteins from other animal species.

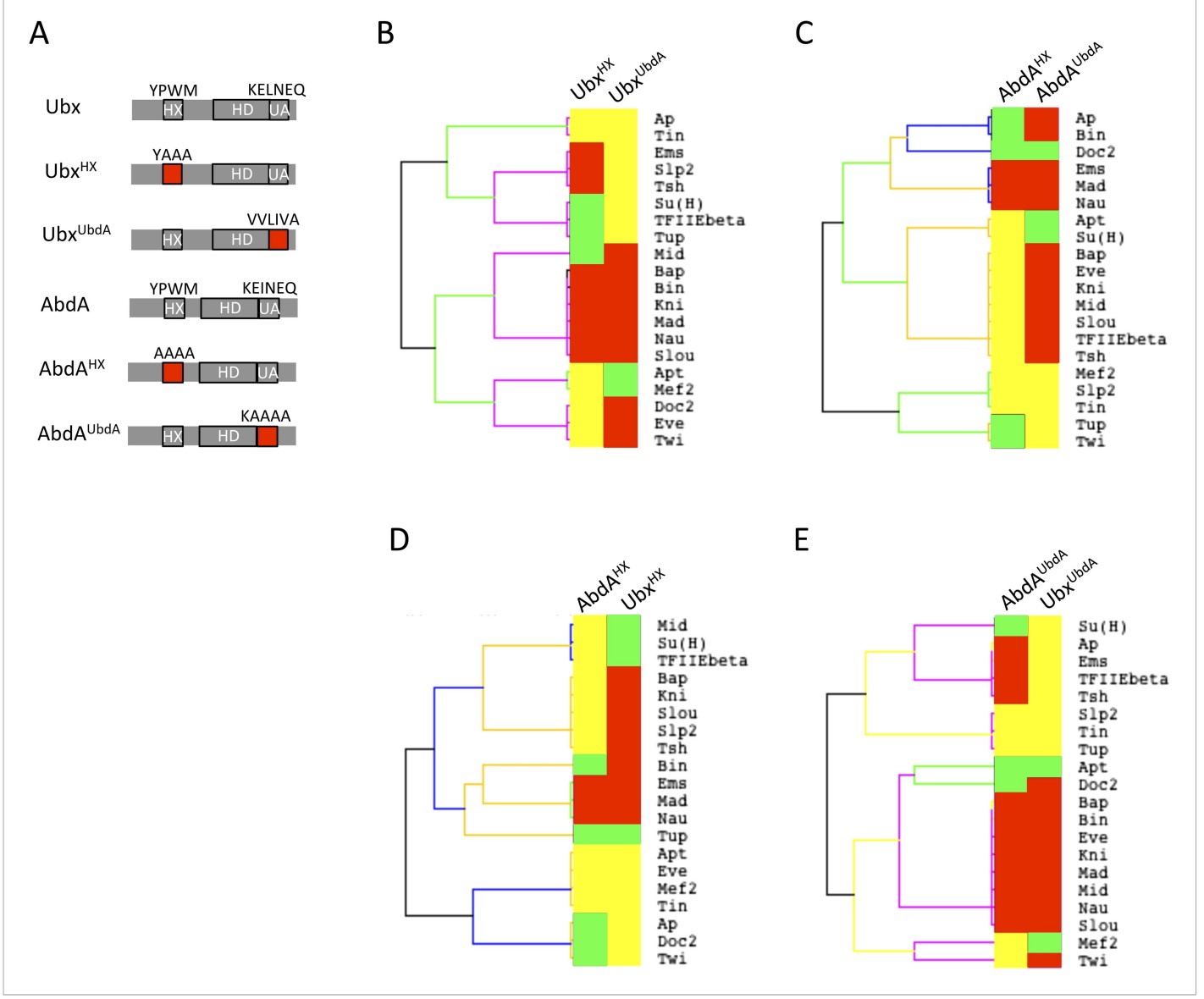

**Figure 6**. Usage mode of the HX and UbdA motifs in Ubx and AbdA proteins. (**A**) Schematic representation of wild type and HX- or UbdA-mutated Ubx and AbdA proteins. (**B**) Heatmap comparing interaction properties of HX- and UbdA-mutated Ubx proteins with a set of 20 TFs. These TFs are common to Ubx and AbdA for BiFC. See also *Figure 6—figure supplement 1*. (**C**) Heatmap comparing interaction properties of HX- and UbdA-mutated AbdA proteins with the same set of TFs. See also *Figure 6—figure supplement 2*. Note that the HX and UbdA mutations have distinct or opposite effects for most of the interactions in Ubx and AbdA. (**D**) Heatmap comparing interaction properties of HX-mutated Ubx and AbdA proteins with the 20 common TFs. (**E**) Heatmap comparing interaction properties of UbdA-mutated Ubx and AbdA proteins with the 20 common TFs. Note that a higher proportion of TFs is similarly affected by the UbdA mutation in Ubx and AbdA when compared to the HX mutation. Colour code is as in *Figure 4*.

The following figure supplements are available for figure 6:

**Figure supplement 1**. BiFC with UbdA-mutated Ubx in the epidermis.

**Figure supplement 2**. BiFC with UbdA-mutated AbdA in the epidermis.

## Inhibition of PPIs is an evolutionary conserved feature of the HX motif

Our work revealed that the HX could specify *Drosophila* Hox interactomes in part by limiting the interaction potential in a context-dependent manner. Here, we ask whether the inhibitory role of the HX on PPIs could constitute an evolutionary conserved property of Hox proteins. To this end, we used

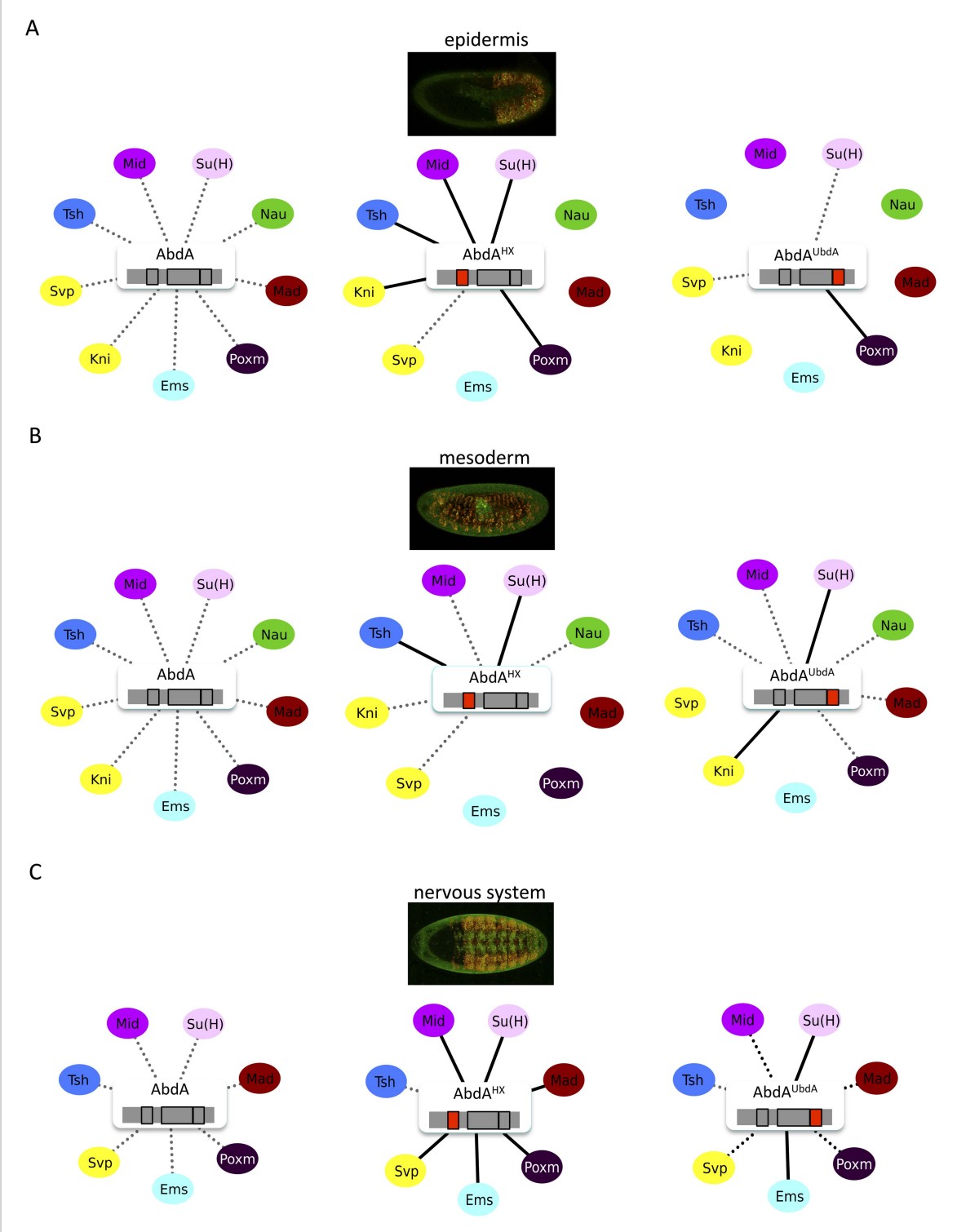

**Figure 7**. The HX and UbdA motifs of AbdA have different interaction properties in different embryonic tissues. (**A**) Interaction properties of wild type and HX- or UbdA-mutated AbdA in the epidermis. (**B**) Interaction properties of wild type and HX- or UbdA-mutated AbdA in the mesoderm. See also *Figure 7—figure supplement 1*. (**C**) Interaction properties of wild type and HX- or UbdA-mutated AbdA in the nervous system. See

*Figure 7. continued on next page*

*Figure 7. Continued*

also *Figure 7—figure supplement 2*. Picture of an embryo making BiFC (green) and expressing the dsRed fluorescent protein under the control of the Gal4 driver illustrates the tissue of interest in each condition. Interactomes are represented as in *Figure 3*.

The following figure supplements are available for figure 7:

**Figure supplement 1**. BiFC with wild type, HX- or UbdA-mutated AbdA in the mesoderm.

**Figure supplement 2**. BiFC with wild type, HX- or UbdA-mutated AbdA in the nervous system.

the mouse HoxB8 and *Nematostella* HoxE proteins as extreme representatives (*Figure 8A*). HoxB8 is a central Hox protein containing a typical HX motif. HoxE was recently shown to display central-like molecular properties (*Hudry et al., 2014*), although it contains a posterior-like derived HX motif (*Figure 8A*). Overall, HoxB8 and HoxE show little sequence identity with the *Drosophila* AbdA protein outside the HX and HD (*Figure 8A*).

We first addressed whether the HoxB8 and HoxE could interact with the 35 *Drosophila* TFs, as previously done with *Drosophila* Hox proteins (see also 'Materials and methods'). Results show that the mouse and cnidarian Hox proteins interact with a surprisingly large number of TFs, respectively 27 and 22 (*Figure 8B–C* and *Figure 8—figure supplements 1, 2*). Still, each Hox protein interacts with a different set of TFs, underlining the existence of preferential/specific interaction properties when considering the whole interactome.

To directly assess whether the HX could act as an inhibitory motif, we analysed the interaction properties of HX-mutated HoxB8 and HoxE proteins, focusing on TFs that were negative with the corresponding wild type Hox proteins. We observed that more than half of the tested TFs became positive with HX-mutated proteins in both cases (*Figure 8—figure supplements 3, 4*). Thus, the HX is also an inhibitory interaction motif in HoxB8 and HoxE.

To further explore how the HX motif could inhibit PPIs, we considered the HoxE protein, which has a simple organisation in terms of secondary structures. Basically, this protein contains a long intrinsically disordered N-terminal region followed by the ordered HD (*Figure 9A*). Still, this protein establishes a number of common interactions with HoxB8 and AbdA (*Figure 8—figure supplement 5*). Since the HD is unlikely to be sufficient for several of those interactions (as deduced from AbdA: *Figure 3C*), we decided to test the N-terminal region of HoxE. We thus generated fly lines carrying a short HoxE variant, called Nter-HoxE, which corresponds to the residues 1 to 54 and does not contain the HX motif (see 'Materials and methods'). BiFC with the 35 *Drosophila* TFs shows that only five interactions are lost when using this short variant of HoxE (*Figure 9B* and *Figure 9—figure supplement 1*). In contrast, this fragment interacts more strongly or establishes new interactions with seven TFs while 15 other interactions remained unaffected. In total, Nter-HoxE establishes as many interactions as the full length HoxE.

Together these results show that the long disordered region of HoxE is involved in a number of the heterologous interactions observed with *Drosophila* TFs. The observation of seven ectopic interactions also underlines that the interaction potential of the disordered region is tightly controlled in the context of the full-length protein. We propose that SLiMs such as the HX motif are important mediators of this control.

## Discussion

Our work revealed a striking propensity of Hox proteins to interact with different types of TFs. In this context, SLiMs such as the HX are not only important for promoting but also for limiting this strong interaction potential. Effects could vary depending on the Hox protein and the tissue considered, highlighting the adaptability of SLiMs to different environments. Thus, the constraining activity of SLiMs on PPIs is an essential attribute of Hox interactome specificity in vivo.

### A novel set of Hox interacting partners in the fly embryo as revealed by BiFC

Apart from the PBC class, very little is known about the TFs that could help Hox proteins to elicit specific developmental programs in the embryo. As a consequence, the interactome underlying Hox

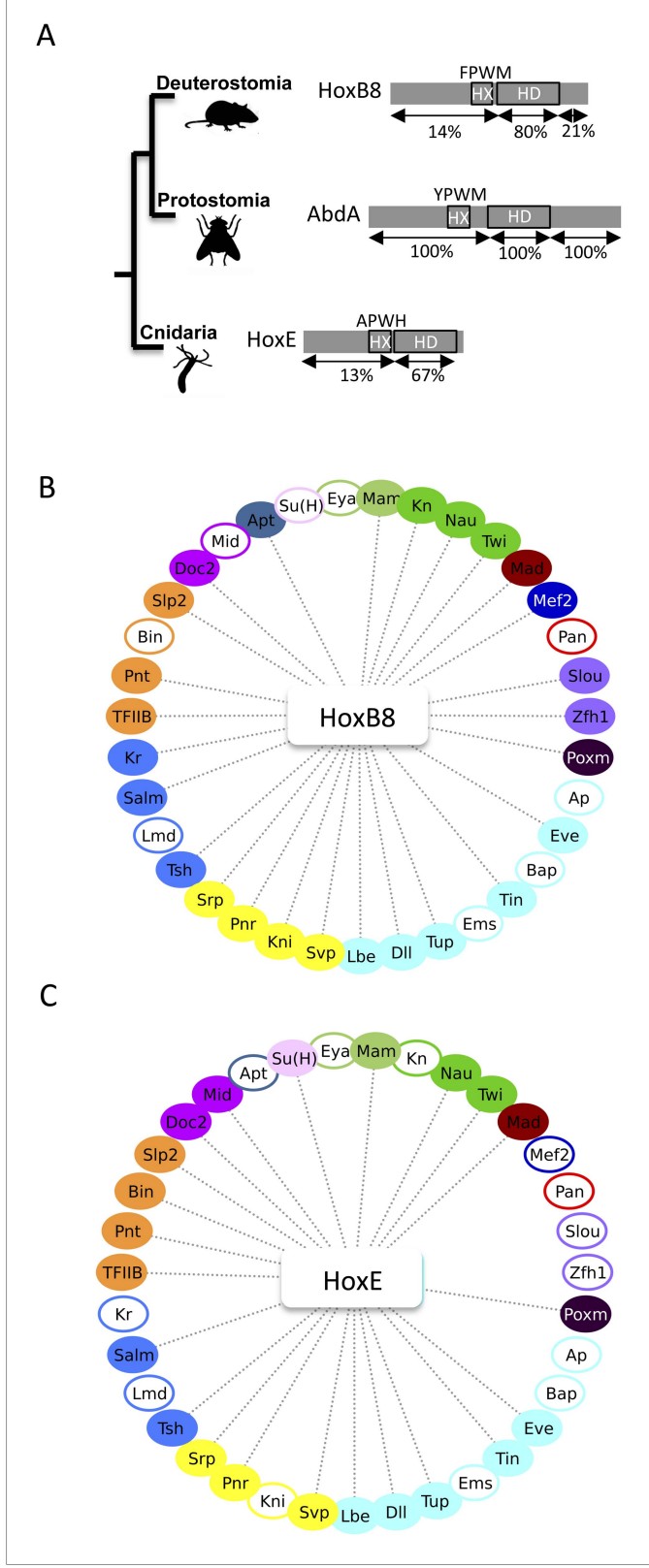

**Figure 8**. The *Nematostella* HoxE and mouse HoxB8 proteins interact with several *Drosophila* TFs. (**A**) Schematic representations of the Hox proteins and the corresponding animal phylogeny. The percentage of sequence identity is given in comparison to AbdA. (**B**) Interactome between mouse HoxB8 and the 35 *Drosophila* TFs. (**C**) Interactome
*Figure 8. continued on next page*

*Figure 8. Continued*

between *Nematostella* HoxE and the 35 *Drosophila* TFs. Colour code and representation are as in *Figure 3*. See also *Figure 8—figure supplements 1–5*.

The following figure supplements are available for figure 8:

**Figure supplement 1**. BiFC between *Drosophila* TFs and the mouse HoxB8 protein.

**Figure supplement 2**. BiFC between *Drosophila* TFs and the *Nematostella* HoxE protein.

**Figure supplement 3**. BiFC between *Drosophila* TFs and HX-mutated HoxB8.

**Figure supplement 4**. BiFC between *Drosophila* TFs and HX-mutated HoxE.

**Figure supplement 5**. *Drosophila* AbdA, mouse HoxB8 and *Nematostella* HoxE interact with several common TFs in vivo.

embryonic functions remains largely elusive. This lack of knowledge is explained by the difficulty of identifying the transcriptional partners that could participate in each context-specific activity of Hox proteins. We hypothesize that many of those interactions are too weak and/or too dynamic to be efficiently trapped through classic high throughput approaches. In support of this, a yeast two-hybrid screen with the *Drosophila* Ubx protein led to the characterization of less than 15 TFs as interacting partners (*Bondos et al., 2006*), strongly contrasting with the numerous functions ensured by this Hox protein during embryogenesis. A similar approach with the mouse HoxA1 protein also led to the characterization of less than 20 TFs (*Lambert et al., 2012*).

Our approach relied on the sequential analysis of tandem interactions between Hox proteins and individual TFs. The interaction screen was performed in two complementary steps with a set of TFs covering different DNA-binding families and displaying various expression profiles during *Drosophila* embryogenesis. This set of TFs is expected to be representative of the diverse transcriptional regulatory activities of Hox proteins in vivo.

The competition experiment was performed in the epidermis, even for TFs that are not endogenously expressed in this tissue. Despite this limitation, we found a high proportion (33/80) of positive competitive events, which confirmed the sensitivity of BiFC for this type of approach. A similar strategy was reported in cell culture for identifying drug molecules that could affect the assembly or the localisation of a specific protein complex (*Morell et al., 2008*). Thus, competitive BiFC could certainly be applied more generally in the future for selecting any kind of new interacting molecules upon the screening of subtle variations in fluorescent reporter signals.

BiFC then showed that all but three competitive TFs could interact with AbdA, making a total of 31 TFs as new Hox interacting partners. In comparison, only seven TFs were so far described to interact with AbdA (*Merabet and Dard, 2014*). These results were reproduced in different tissues of the embryo or corroborated by co-ip experiments in S2 cells, illustrating that Hox-TF interactions could occur in different cell contexts upon co-expression. Importantly, the specificity of each interaction was supported by the loss of fluorescent signals when using mutated or truncated Hox variants, and by the observation of typical nuclear interaction profiles with different TFs.

It is important to stress that all these results revealed an interaction potential between Hox proteins and TFs. Whether and how these interactions could be used in the context of the endogenous gene products is an open question. For example, some of the interactions revealed with the fusion Scr protein are unlikely to occur with the endogenous Scr product, which displays a quite restrained expression profile in the embryo compared to the other tested Hox proteins (*Supplementary file 3* and [*Hammonds et al., 2013*]). In addition, the fact that TF-encoding genes were not expressed under the control of their endogenous promoter forbids assessing the role of tissue-specific expression levels in Hox interactome properties. In this context, the recent advent of genetic tools in *Drosophila*, including Mimic elements (*Gnerer et al., 2015*) and the CRISPR/Cas9 system (*Bassett and Liu, 2014*), could certainly add to the functional relevance of BiFC observations in the future. Nevertheless, the high proportion of positive events among a starting set of 80 TFs strongly suggests that AbdA, and

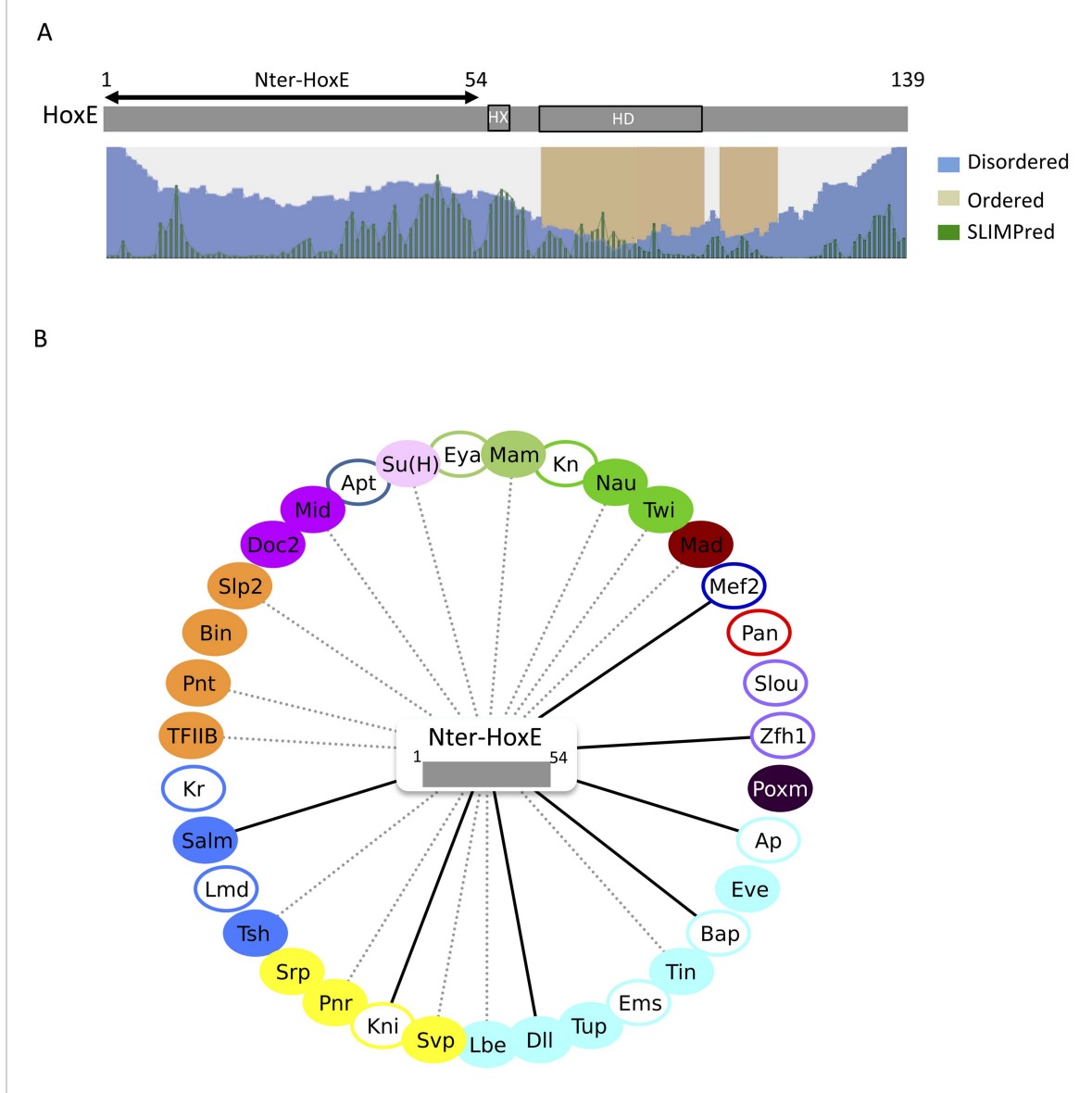

**Figure 9**. The intrinsically disordered region of HoxE establishes a number of interactions with *Drosophila* TFs. (**A**) Scheme of full length *Nematostella* HoxE with its predicted SLiMs (green bars), and disordered (blue waves) or ordered (brown blocks) regions. Adapted from iupred (http://iupred.enzim.hu/). The N-terminal disordered region used for BiFC is indicated (Nter-HoxE). (**B**) Interactome between Nter-HoxE and the 35 *Drosophila* TFs. Colour code and representation are as in *Figure 3*. See also *Figure 9—figure supplement 1*.

The following figure supplement is available for figure 9:

**Figure supplement 1**. BiFC between *Drosophila* TFs and the N-terminal (1-54) fragment of the *Nematostella* HoxE protein.

Hox proteins in general, have a strong potential to interact with a number of different TFs in vivo. This assumption could be verified by using a library of 600 TFs compatible for BiFC in *Drosophila* (*Bischof et al., 2013*).

## Specific vs common cofactors among Hox interactomes

Hox proteins play numerous functions in all embryonic germ layers. These functions can be highly specific (*Brodu et al., 2002*; *Li-kroeger et al., 2008*) or common to several (*Gebelein et al., 2002*; *Coiffier et al., 2008*) Hox proteins, suggesting they could rely on the interaction with different types of cofactors.

Here, we present the first interactomes of five Hox proteins with a set of 35 TFs. BiFC and co-ip experiments revealed that all tested TFs could interact with two or more Hox proteins. Although this result should be expanded to more TFs, it suggests that the interaction between Hox proteins and TFs is generally not exclusive. As a corollary, we hypothesize that Hox interactome specificity is unlikely to rely solely on the interaction with specific TFs.

Despite a number of common interactions, each Hox interactome contains a different set of positive TFs. AbdA is the Hox protein establishing the highest number of interactions, which is consistent with the fact that it served as a bait protein in the starting competition screen. However, the observation that Hox proteins do not interact systematically with the same set of cofactors shows their specificity. Interestingly, this specificity is not only occurring at the DNA-binding level since the loss of AbdA DNA-binding activity did not affect all interactions (18 interactions of 31 were affected). Our results thus emphasize the need of considering post-DNA-binding mechanisms for understanding Hox functional specificity in vivo.

The strong interaction potential of Hox proteins is consistent with their wide spectrum of regulatory activities in the embryo. It probably constitutes an inherent feature of other classes of developmental TFs that intervene in several regulatory processes throughout embryogenesis. Along this line, observations from genome-wide analyses showed that target cis-regulatory sequences allow the assembly of large multi-protein complexes (*Moorman et al., 2006*; *Blaxter, 2010*; *Kvon et al., 2012*). In addition, TFs can generally bind thousands of sites across the genome (*Li et al., 2008*). Thus, the high number of possibilities in protein–protein and protein-DNA contacts likely reflects the propensity of developmental TFs to regulate target gene expression in many different cell contexts.

Finally, clustering analyses showed that the similarity between the different Hox interactomes does not follow the level of sequence identity between Hox proteins. Thus, common interactions might rely on different Hox protein interfaces. Accordingly, we found that the HD, which contains the highest score of sequence identity between Hox proteins, was not sufficient to ensure the majority of interactions established by AbdA. In addition, the common HX motif is not similarly used in the different Hox interactomes. Interaction properties of the HX motif were also different depending on the tissue considered, highlighting the strong flexibility and adaptability of this motif to the surrounding protein environment. Restricting the analysis to the closely related Ubx and AbdA proteins did not reveal a higher level of similarity in the interaction properties of the HX motif. In contrast, the UbdA motif showed a more frequent similar usage mode between the two Hox proteins. Thus, SLiMs conserved at different evolutionary extents provide different levels of specificity to Hox interaction properties.

## Evolutionary perspective of Hox interactomes

A high interaction potential was not only observed with *Drosophila*, but also with mouse and cnidarian Hox proteins. This result is particularly striking with the cnidarian HoxE protein, which is capable of interacting with TFs that are specific of the Bilateria group, including Biniou, Midline, Pointed or Teashirt. Although these observations are not functionally informative, they indicate that the strong interaction potential of Hox proteins is an ancestral feature that was probably present before its full exploitation in bilaterian lineages.

In addition, the observation that highly divergent Hox proteins could interact with the same set of cofactors questions the role of conserved and non-conserved regions in Hox functions. Interestingly, long intrinsic disordered regions characterize Hox proteins in general (*Merabet and Dard, 2014*), and a recent study showed that they could serve in Ubx to bind different partners in a competitive or cooperative way (*Hsiao et al., 2014*). Results obtained with the cnidarian HoxE protein confirm the important role of a long disordered region in mediating interactions with different TFs. We suggest that the acquisition of long intrinsic disordered regions was a key for providing functional diversity to Hox proteins during animal evolution.

## A revised view of SLiMs in mediating protein–protein interactions

Our work provides an original experimental strategy for analysing the role of SLiMs in the context of full-length proteins in vivo. Results show that the HX mutation affects a number of interactions in all tested Hox proteins. Surprisingly, the absence of the HX motif could lead to a stronger or new interaction potential with TFs. A gain of interaction was observed with *Drosophila*, mouse, and cnidarian Hox proteins, suggesting that this molecular property is evolutionary conserved in the

animal kingdom. In total, the HX motif appears more often involved in limiting rather than in promoting interactions with TFs.

The inhibitory effect of the HX motif on PPIs was most pronounced in Ubx and AbdA, which were also used for the analysis of the UbdA motif. We found that this motif was also required for limiting the interaction potential of the two Hox proteins, although to a lesser extent than the HX motif. Thus, the negative regulation of PPIs is not a specific property of the HX motif.

Inhibitory activity of SLiMs on Hox protein function can be reconsidered in the light of previous functional data. For example, the HX mutation was shown to convert AbdA from a repressor to a strong activator of the *decapentaplegic* (*dpp*) target enhancer (*Merabet et al., 2011*). This striking transcriptional conversion is difficult to assign to a simple loss of interaction with co-repressor(s). Along the same line, the HX mutation increases the interaction potential of Ubx with Exd in vivo (*Hudry et al., 2012*), and confers an AbdA-like activity to Ubx for segment specification in the epidermis (*Galant et al., 2002*). Similarly, the mouse HoxB8 protein was shown to provoke dominant negative phenotypes in absence of its HX motif (*Medina-martinez & Ramı, 2003*), which is also difficult to reconcile with a simple loss of interactions. Finally, the interaction between HoxA11 and Foxo1a in placental mammal lineages is reported to result from the loss of a specific Foxo1a–inhibitory interaction domain and not from the gain of a new binding interface in HoxA11 (*Brayer et al., 2011*). The gain of interaction between HoxA11 and Foxo1a comprises one of the major regulatory events that led to placentation in mammals, illustrating a so far unexpected role of a PPI inhibitory domain in interactome rewiring during evolution (*Lynch et al., 2008*).

More generally, protein autoinhibition is described for the regulation of other molecular events, including protein-DNA interactions and actin polymerisation (*Pufall, 2002*; *Lee et al., 2005*; *Padrick and Rosen, 2010*). It relies on the presence of inhibitory modules that were recently found to be enriched in intrinsically disordered regions (*Trudeau et al., 2013*). SLiMs can also be categorized as hiding motifs (*Van Roey et al., 2013*), but this role has so far been described for few mammalian proteins in the context of their intracellular transport or post-translational modification.

Here, we demonstrate that two different SLiMs could be used in a cell type-dependent manner for promoting or limiting the Hox interaction potential with TFs. We propose different models for explaining the underlying molecular mechanisms. In the classic situation, SLiMs are positively used as context-specific interaction modules, together with globular domains, for the recruitment of cell-specific cofactors (cell context 1 in *Figure 10*). Alternatively, SLiMs could also be important for restraining the interaction potential (cell contexts 2 and 3 in *Figure 10*). In one mechanism, the inhibitory activity would rely on the interaction with a particular partner that will mask or forbid the recruitment of the other SLiM-interacting cofactors (cell context 2 in *Figure 10*). This mechanism implies that the interaction between the SLiM and the hiding partner will be strong and stable enough to overcome the binding of the other cofactors. In a second mechanism, SLiMs could also directly act as a masking peptide, preventing recognition and/or binding of undesired cofactors (cell context 3 in *Figure 10*). SLiMs are classically required to make interactions with structured globular domains in *trans* (*Stein and Aloy, 2008*), but intramolecular contacts following post-translational modifications have also been reported in a few cases (*Pawson et al., 2001*). Thus, SLiMs could also establish interactions in *cis*, potentially upon cell-specific modifications (i.e., phosphorylations), to eventually inhibit the recruitment of inappropriate cofactors. A similar role was previously described for the HX motif of Labial, which is able to prevent the binding of the HD on the DNA (*Chan et al., 1996*). Interestingly, this inhibition is relieved upon interaction with Exd, highlighting the influence of the environment on the SLiM inhibitory activity.

In summary, we showed that highly conserved SLiMs are used in a context-dependent manner for constraining the interaction potential of Hox proteins with surrounding TFs. This molecular strategy has certainly been underestimated to date. We propose that the inhibiting interaction properties of SLiMs could apply more generally to the fine-tuning of highly connected interactomes. It is interesting to note that SLiMs can also be produced as individual molecules from short open reading frames (*Kondo et al., 2007*; *Magny et al., 2013*) or even from long non coding RNAs (*Ruiz-Orera et al., 2014*). We anticipate that short peptides could act as buffering molecules, helping hub proteins to discriminate their correct partners among hundreds of possible interactions within the 'messy' (*Tawfik, 2010*) cell environment.

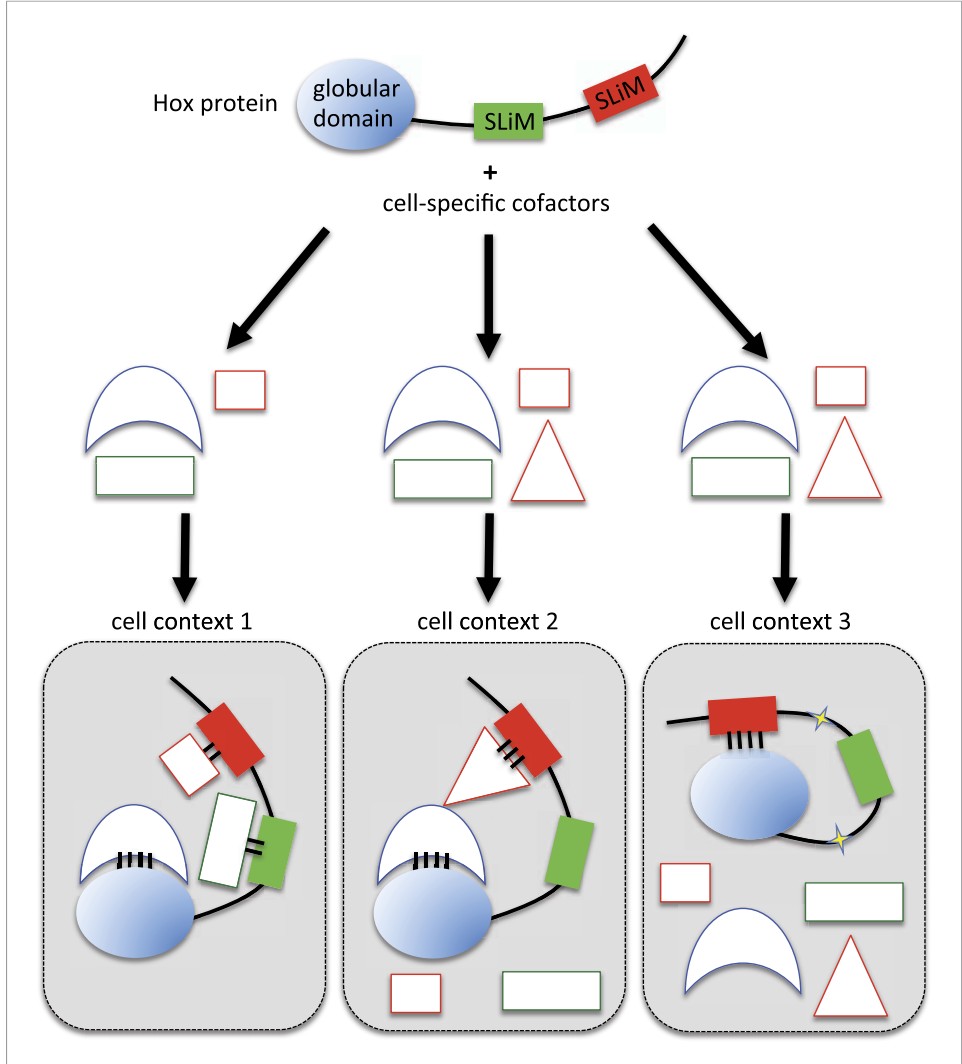

**Figure 10**. Molecular mechanisms underlying context-dependent activities of SLiMs in protein–protein interactions. The Hox protein is represented as containing a globular structured domain together with two different SLiMs embedded in a disordered region, as indicated. This protein will present different interaction properties with a set of cofactors that could vary depending on the cell context considered. Preferential interactions between cofactors and the protein domain and SLiMs are represented by a colour code. Black bars symbolize the various levels of interaction affinity. In the cell context 1, cofactors are recruited through specific interactions with the globular domain and the two SLiMs. In the cell context 2, there is a supplementary triangular cofactor that displays higher affinity with the red SLiM than the square cofactor. As a consequence, interaction will occur with this triangular (hiding) cofactor, which forbids the interaction with the other SLiM. In this context, the red SLiM behaves as an inhibitory interaction motif. In the cell context 3, post-translational modifications in the disordered region (yellow stars) allow the inhibitory SLiM to establish interactions in *cis* with the globular domain. These intra-molecular contacts forbid the binding of the other cofactors. The last two mechanisms illustrate how the inhibitory activity of SLiMs could help in distinguishing/specifying interactomes with an identical set of cofactors.

# Materials and methods

## Fusion protein constructs and transgenic lines

Several VC-Hox fusion constructs were previously generated (*Hudry et al., 2012*; *Boube et al., 2014*): these correspond to wild type and mutated/truncated variants of all *Drosophila* Hox proteins and wild type and HX-mutated HoxE. Other Hox fusion constructs (HoxB8, HoxB8[HX] and Nter-HoxE) were generated by PCR and restriction-cloned in the pUAST or pUASTattB vector, in fusion with the C-

terminal (155–238) fragment of Venus (VC) at 3′ end (see *Supplementary file 4*). Fusion TFs were also generated by PCR from full-length complementary DNAs and restriction-cloned in fusion in pUAST or pUASTattB vector with the N-terminal (1–173) fragment of Venus (VN) at the 5′ or 3′ end (see *Supplementary file 4*). Primers used for cloning of each TF are listed in *Supplementary file 5*. For all fusion constructs, a linker of three to five amino acids was added to separate the Venus fragment from the protein. All constructs were sequence-verified before injection. Transgenic lines were established either by the PhiC31 integrase system (with the pUASTattB vector [*Venken et al., 2006*; *Bischof et al., 2007*]) or by classic P-element (with PUAST vector) mediated germ line transformation. Expression level of Hox fusion constructs was verified as previously described (*Hudry et al., 2011*). Briefly, flies were crossed at different temperatures with the *abdA-Gal4* driver and embryos were collected for immunostaining with a chicken anti-GFP antibody (Abcam ab13970, England) recognizing the VN and VC fragments. Fluorescent revelation (with a secondary anti-chicken antibody coupled to AlexaFluor488, Abcam150169) was used to compare the expression level between the different conditions with wild type and mutated Hox fusion constructs. The temperature for each fly cross was adjusted accordingly, allowing comparing BiFC signals with Hox fusion proteins expressed at similar levels. The same anti-GFP antibody was used to verify the correct expression level of each generated VN-TF fly line.

## Fly stocks and genetic crosses

Gal4 drivers used are: *Antp-Gal4*, *Ubx-Gal4*, *abdA-Gal4*, and *AbdB-Gal4* (*de Navas et al., 2006*; *Hudry et al., 2012*). Fly lines generating in this work are listed in *Supplementary file 4*.

Competition tests were performed in one generation by crossing each candidate *UAS-TF* (see *Supplementary file 1* for the type of the UAS fly line) with the BiFC reporter fly line containing the *UAS-VC-abdA* (homozygous on the second chromosome) and *UAS-VN-exd* (homozygous on the fourth chromosome) constructs, together with the *abdA-Gal4* driver balanced over a *TM6tubulineGal80* third balancer (*Hudry et al., 2011*). Under these conditions, half of the embryo progeny (with homozygous UAS-TF fly line) could display affected BiFC signals in presence of a competitive TF.

Complementation tests were performed by crossing *en mass* virgin females containing the VN-TF and VC-Hox constructs (as non-established fly lines resulting from a previous cross between VN-TF and VC-Hox individuals) with males carrying the corresponding *Hox-Gal4* driver. Over night egg laying was performed at different temperatures, according to the expression level of the VC-Hox variant.

## Immunostaining

Embryo collection, preparation, and immunodetections were performed according to standard procedures. The antibodies used were: chicken anti-GFP (Abcam ab13970, 1/500), mouse anti-Scr (6H4.1, 1/100), mouse anti-Antp (4C3, 1/100), mouse anti-Ubx (FP3.38, 1/100), rabbit anti-AbdA (Dm. Abd-A.1, 1/100), mouse anti-AbdB

## Co-immunoprecipitation (Co-ip) experiments and western blot

Co-ips were performed in S2 cells, which were transfected with a HA-tagged form of AbdA, together with an actin-Gal4 plasmid and the corresponding VN-cofactor. AbdA-HA construct was generated by PCR, using an oligonucleotide bearing the HA sequence and cloned into the pUASTattB vector. The construct was sequence verified. Transfection was realised using the X-tremeGENE HP DNA Transfection Reagent (Roche). Cells were lysed 48 hr later and nuclear extracts were prepared as classically described. Ip was performed with a polyclonal rabbit anti-HA antibody (Abcam ab9110). Presence of AbdA-HA was verified by western blot using a monoclonal anti-HA antibody (HA.11 from Covance). Presence of the associated VN-cofactor was revealed with a chicken anti-GFP antibody recognising the VN fragment (Abcam ab13970).

## BiFC analysis in *Drosophila* embryos

Experimental parameters allowing a comparable expression level between wild type and mutated VC-Hox proteins were previously established for several constructs (*Hudry et al., 2012*; *Boube et al., 2014*) or deduced from additional immunostaining experiments for new constructs (HoxB8, HoxB8[HX], and Nter-HoxE). Fly crosses for BiFC analyses were set up at the defined temperature over night.

After the removal of the flies, the embryos were kept at 4°C for 24 hr before live imaging. Live embryos were dechorionated and mounted in the halocarbon oil 10S (commercialized by VWR, Pennsylvania, USA). Quantification of the BiFC signals was realised by taking unsaturated images at the same desired stage, depending on the tissue considered (epidermis: stage 10, mesoderm: stage 12, nervous system: stage 14). For BiFC analysis in the CNS, embryos were manually aligned on the dorsal face. Observations were performed at least twice (from two different over night egg laying periods) for a same genotype. A minimum of 5 embryos of the correct developmental stage was considered in each case. Pictures were acquired using a LSM780 confocal microscope (Zeiss, Jena, Germany). For Venus fluorescence, filters were adjusted at 500 nm for excitation and 535 nm for emission. Identical parameters of acquisition were applied between the different genotypes. The number and intensity of the all pixels (for each embryo) were measured in the tissue of interest using the histogram function of the ImageJ Software. The quantification of fluorescence complementation is shown for each condition by boxplot representation using R-Software. Boxplot depicts: the smallest value, lower quartile, median (black line), upper quartile, and largest value for each condition.

### Network visualisation and heatmap
Networks were represented using Cytoscape 3.0 (*Shannon et al., 2003*). A hierarchical clustering algorithm (with Euclidian distance and average linking) was applied to the matrix using the MeV software suite (*Saeed et al., 2006*). The bootstrap method was used for resampling the data and provides a statistical support for each tree node.

## Acknowledgements

We thank Johannes Bischof and Bart Desplancke for sharing unpublished tools. We thank Muriel Boube, Henri-Marc Bourbon, and Luis Mojica-Vasquez for the *Distalless-VN* fly line, Damien Ahr for his help in fly genetics and BiFC observations and Yacine Graba for freedom during the earliest part of this work in his laboratory. We thank Eileen Furlong, Christophe Jagla, Manfred Frasch, Marc Haenlin, Michael Taylor, Alan M. Michelson, Stefan Thor, James Castelli-Gair, the Bloomingon Stock Center and FlyORF for fly lines and cDNAs. We warmly thank François Payre, Andreas Zanzoni, and Johannes Bischof for their helpful comments on the manuscript and Joanne Burden for proofreading. This work was supported by grants from ARC and FRM.

## Additional information

### Funding

| Funder | Author |
| --- | --- |
| Fondation pour la Recherche Médicale | Séverine Viala, Marjorie Heim, Amélie Dard, Bruno Hudry, Marilyne Duffraisse, Ana Rogulja-Ortmann, Christine Brun, Samir Merabet |
| Association pour la Recherche sur le Cancer | Séverine Viala, Marjorie Heim, Amélie Dard, Bruno Hudry, Marilyne Duffraisse, Ana Rogulja-Ortmann, Christine Brun, Samir Merabet |

The funders had no role in study design, data collection and interpretation, or the decision to submit the work for publication.

### Author contributions

MB, SV, MH, AD, BH, MD, AR-O, Acquisition of data, Analysis and interpretation of data; CB, Analysis and interpretation of data, Drafting or revising the article; SM, Conception and design, Acquisition of data, Analysis and interpretation of data, Drafting or revising the article, Contributed unpublished essential data or reagents

### Author ORCIDs

Christine Brun, http://orcid.org/0000-0002-5563-6765

# Additional files

## Supplementary files

• Supplementary file 1. List of the 80 candidate transcription factors (TFs) tested as potential cofactors of AbdA. Each TF used in the competition test corresponds either to an artificial UAS construct or to a UAS-containing transposon inserted in the endogenous promoter locus. These constructs were obtained from stock centres or from particular laboratories, as indicated. Red and green-filled boxes indicate positive or negative competitor status, respectively. The expression profile of each TF is provided for the main embryonic germ layers and was compiled according to databases (http://flybase.org and [*Hammonds et al., 2013*]). Yellow and black-filled boxes depict presence or absence of expression, respectively. TFs that were subsequently used for BiFC with AbdA are bolded and annotated with a grey box. Note that TFIIbeta was directly included in the BiFC analysis without doing a preliminary competition test (see main text for details).

• Supplementary file 2. Spatiotemporal expression pattern of AbdA and TFs under study as deduced from in situ hybridization experiments (*Hammonds et al., 2013*).

• Supplementary file 3. List of the TFs used for BiFC in the mesoderm. Colour code for tissue expression is as in *Supplementary file 1*. Green or red boxes depict presence or absence of BiFC, respectively. Results obtained by co-ip in S2 cells are also indicated for each TF. Note that only Kn is interacting specifically in the mesoderm among the tested TFs.

• Supplementary file 4. List of the constructs and transgenic fly lines generated in this study.

• Supplementary file 5. List of the primers used to clone each TF in fusion with the Venus fragment.

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
