## [Decision Letter]

Thank you for sending your work entitled “Inhibitory activities of short linear motifs underlie Hox interactome specificity in vivo” for consideration at *eLife*. Your article has been favorably evaluated by K VijayRaghavan (Senior editor) and two reviewers, one of whom is a member of our Board of Reviewing Editors.

The Reviewing editor and the other reviewer discussed their comments before we reached this decision, and the Reviewing editor has assembled the following comments to help you prepare a revised submission.

This is a very data-rich paper describing an impressive number of experiments to systematically elucidate interactions of a range of transcription factors with Hox proteins.

The paper has the character of a screening exercise, complemented with a few experiments to further elucidate mechanisms. However, it does not go much in depth with respect to mechanisms (i.e. actual definition of interaction motifs). Its strength is in showing a surprising number of seemingly complex and tissue-dependent interactions between transcription factors and Hox proteins.

However, one may be concerned about how many of these interactions are relevant to the functioning of the particular Hox proteins in vivo in their endogenous roles. But regardless of this potential criticism of the approach, the research does very clearly show the huge potential for a very wide range of interactions, which has important evolutionary implications even if a specific interaction is not happening in vivo in *Drosophila* at this particular point in evolutionary time.

Still, it would be prudent to be very self-critical about the data and discuss clearly that it is open which part of the interactions may be truly functional. It seems particularly worrying that even highly diverged Hox proteins from very distant taxa can be used in the assay and interactions with *Drosophila* transcription factors are retained. One might suspect that one could use any intrinsically disordered protein region and show similar interactions. While it would not seem reasonable to add a lot of experiments at this point to address this question, an extended self-critical discussion would be required (i.e., extending the self-critical assessment throughout the manuscript and adding also a word of caution into the Abstract). It is predictable that this paper will be controversially discussed, but it could also be a paper that opens a new field.

Minor comments:

My only concerns or criticisms are about making the manuscript as easy to read and understand as possible.

In the Results, the description of the logic behind the experimental design (even with Figure 1) was not very clear to me. I found myself getting confused and having to re-read the early parts of the Results sections several times. More specifically, it was the logic behind the competitive BiFC that was not entirely clear. Third paragraph: why “independently”? Interactions of partners/co-factors could also be “dependent” on an AbdA-Exd complex.

The titration effect between AbdA and Exd when cold AbdA or Exd is introduced into the system is clear, as one of the VN-bearing partners is simply being displaced from the complex. However, adding other cold “partners” that are supposed to interact with the AbdA-Exd complex independently does not necessarily disrupt the AbdA-Exd complex unless this new cold partner interacts with sites in between the AbdA and Exd interface zone (?). Thus this partner interaction is not independent of the AbdA-Exd interaction in this context either.

Furthermore, will the approach miss partners/co-factors that do not interact with either AbdA or Exd in regions that are in the AbdA-Exd interaction zone/interface?

I think that one further aspect of the writing that led to my confusion was the way that the approach is initially framed in the context of competitive BiFC with the AbdA-Exd complex, but then the results almost immediately seem to be talking about partners only of AbdA, rather than of partners interacting with the AbdA-Exd complex. I realised what was going on eventually, but perhaps clearer 'sign-posting' of the different experimental approaches and the distinct sections of competitive BiFC versus simply interactions with AbdA would help the reader.

---

## [Author Response]

*Still, it would be prudent to be very self-critical about the data and discuss clearly that it is open which part of the interactions may be truly functional. It seems particularly worrying that even highly diverged Hox proteins from very distant taxa can be used in the assay and interactions with* Drosophila *transcription factors are retained. One might suspect that one could use any intrinsically disordered protein region and show similar interactions. While it would not seem reasonable to add a lot of experiments at this point to address this question, an extended self-critical discussion would be required (i.e., extending the self-critical assessment throughout the manuscript and adding also a word of caution into the Abstract)*.

This point was only discussed in the third paragraph of the Discussion in the previous version. We did not emphasize a self-criticism further since our purpose was to assess the interaction potential of Hox proteins with transcription factors. In addition, understanding the functional contribution of the newly identified interactions will constitute a real project on its own and will go beyond the framework of our message. Nevertheless, we fully agree that more caution should be taken regarding the functional relevance of our data. This concern is particularly important given that BiFC analyses were not performed with the endogenous promoter of TF-encoding genes. BiFC could certainly be improved in the future thanks to the advent of powerful genetic tools (Mimic elements and Crispr technology for example) that now allow generating endogenous fusion proteins in a more systematic and efficient way.

To follow the reviewers’ advice, we have now mentioned the functional limitation of our observations in several places throughout the manuscript, including:

In the Abstract: “Although these interactions remain to be analysed in the context of endogenous Hox regulatory activities, our observations…” We also deleted the last concluding sentence “hence function in vivo”.

In the Results section: “However, whether and how these binding partners could be used in the context of endogenous regulatory activities of AbdA remain to be investigated.”

In the subsection “A novel set of Hox interacting partners in the fly embryo as revealed by BiFC”, in the Discussion: “It is important to stress that all these results revealed an interaction potential between Hox proteins and TFs. […] the recent advent of genetic tools in *Drosophila*, including Mimic elements (Gnerer, 2015) and the CRISPR/Cas9 system (Bassett, 2014) could certainly add to the functional relevance of BiFC observations in the future.”

In the subsection headed “Evolutionary perspective of Hox interactomes”, also in the Discussion: “This result is particularly striking with the cnidarian HoxE protein, which is capable of interacting with TFs that are specific of the Bilateria group, including Biniou, Midline, Pointed or Teashirt. Although these observations are not functionally informative, they indicate that the strong interaction potential of Hox proteins is an ancestral feature that was probably present before its full exploitation in bilaterian lineages.”

Minor comments:

*My only concerns or criticisms are about making the manuscript as easy to read and understand as possible*.

*In the Results, the description of the logic behind the experimental design (even with*
Figure 1*) was not very clear to me. I found myself getting confused and having to re-read the early parts of the Results sections several times. More specifically, it was the logic behind the competitive BiFC that was not entirely clear. Third paragraph: why “independently”? Interactions of partners/co-factors could also be “dependent” on an AbdA-Exd complex*.

The titration effect between AbdA and Exd when cold AbdA or Exd is introduced into the system is clear, as one of the VN-bearing partners is simply being displaced from the complex. However, adding other cold “partners” that are supposed to interact with the AbdA-Exd complex independently does not necessarily disrupt the AbdA-Exd complex unless this new cold partner interacts with sites in between the AbdA and Exd interface zone (?). Thus this partner interaction is not independent of the AbdA-Exd interaction in this context either.

Furthermore, will the approach miss partners/co-factors that do not interact with either AbdA or Exd in regions that are in the AbdA-Exd interaction zone/interface?

*I think that one further aspect of the writing that led to my confusion was the way that the approach is initially framed in the context of competitive BiFC with the AbdA-Exd complex, but then the results almost immediately seem to be talking about partners only of AbdA, rather than of partners interacting with the AbdA-Exd complex. I realised what was going on eventually, but perhaps clearer 'sign-posting' of the different experimental approaches and the distinct sections of competitive BiFC versus simply interactions with AbdA would help the reader*.

We do not consider this critical point as a “minor” comment and we thank the reviewer for his/her confusion. We realize that the way we introduced the competitive BiFC for pre-selecting putative AbdA-interacting partners was indeed not sufficiently clear. Still, this introducing part is important for the understanding of our experimental strategy, which could serve in the future for other studies aiming at rapidly selecting candidate interacting partners. We thus believe that competitive BiFC should be considered as an integrant part of our approach. To be less confusing, we completely rephrased the corresponding paragraphs in the first and second sections of the Results and depicted all the different theoretical cases for non-competitive and competitive BiFC in a new Figure 1 (Figure 1). The word “independent” was no longer used since it could be misinterpreted. We added more information about the limitation of the approach to emphasize the type of interactions that could be or could not be trapped by competitive BiFC. Finally, everything was directly linked to the main objective of finding putative AbdA interacting partners for better clarity.